# Inferring stochastic dynamics with growth from cross-sectional data

**Stephen Zhang**[†]
School of Mathematics and Statistics,
University of Melbourne

**Suryanarayana Maddu**
Center for Computational Biology,
Flatiron Institute

**Xiaojie Qiu**
Department of Genetics,
Stanford University School of Medicine

**Victor Chardès**[†]
Center for Computational Biology,
Flatiron Institute

## Abstract

Time-resolved single-cell omics data offers high-throughput, genome-wide measurements of cellular states, which are instrumental to reverse-engineer the processes underpinning cell fate. Such technologies are inherently destructive, allowing only cross-sectional measurements of the underlying stochastic dynamical system. Furthermore, cells may divide or die in addition to changing their molecular state. Collectively these present a major challenge to inferring realistic biophysical models. We present a novel approach, *unbalanced* probability flow inference, that addresses this challenge for biological processes modelled as stochastic dynamics with growth. By leveraging a Lagrangian formulation of the Fokker-Planck equation, our method accurately disentangles drift from intrinsic noise and growth. We showcase the applicability of our approach through evaluation on a range of simulated and real single-cell RNA-seq datasets. Comparing to several existing methods, we find our method achieves higher accuracy while enjoying a simple two-step training scheme.

## 1 Introduction

Single-cell measurement technologies offer an unbiased, single-cell resolution view of the molecular processes dictating cell fate. These methods predominantly rely on microfluidic devices to capture and read molecules in individual cells, destroying the cell in the measurement process. Therefore the state of any cell can be measured only at a single time. Experimental study of temporal biological processes must therefore proceed by time-course studies, in which a time-series of population snapshots is obtained. At each time-point, a measurement is obtained from a distinct population of cells, assumed to be identically prepared at the initial time. In practice these data can be obtained by serial sampling from a larger population, or from biological replicates measured at different times [1, 2]. The challenge with time-resolved population snapshot data is that direct observation of dynamics in terms of longitudinal information is lost, and must be reconstructed from static population profiles. This motivates the inverse problem of reconstructing an underlying dynamical system which not only interpolates the data, but also captures the key biophysical phenomenon at play in the regulation of gene expression, notably intrinsic noise and cellular proliferation. On the other hand, ignoring these aspects can result in incorrect identification of regulatory mechanism [3, 4].

While the task of learning stochastic dynamics interpolating high-dimensional distributions is now routinely solved in generative modelling tasks [5, 6], doing so while ensuring faithfulness to biophysically relevant features like intrinsic molecular noise as well as division and death remains a

---

[†]Corresponding authors: `syz@syz.id.au`, `vchardes@flatironinstitute.org`.

39th Conference on Neural Information Processing Systems (NeurIPS 2025).

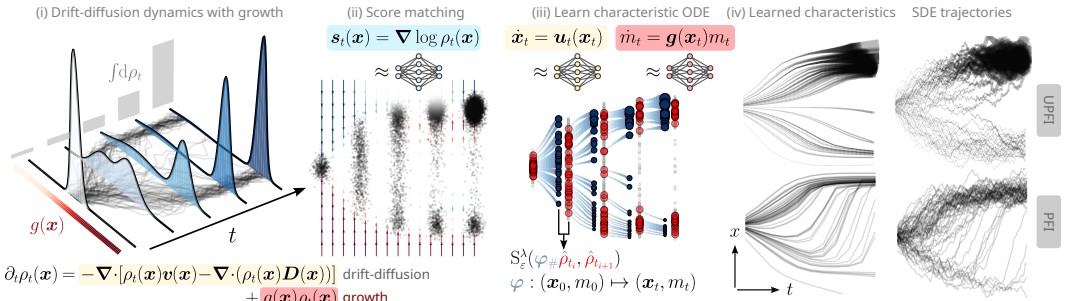

Figure 1: **Overview.** (i) Stochastic population dynamics with growth, governed by a Fokker-Planck equation with source. (ii) Score matching trains a neural network to model the contribution of noise. (iii) Neural-ODE based learning of the Fokker-Planck characteristics. (iv) Learned characteristics and corresponding SDE trajectories.

challenge. Similarly, while the dynamical inference problem has seen significant interest from the single cell analysis community, the issues of the noise and growth remain open for the most part [3, 4, 7, 8]. For instance, most existing methods assume either no noise [9] or a constant isotropic diffusivity [2, 10], and either no growth [11] or the availability of prior information such as growth rate estimates [2] or lineage tracing [4, 12].

With this in mind, a flexible framework called *probability flow inference* (PFI) was recently introduced [13, 14] as a tool to infer stochastic dynamical models accounting for intrinsic noise. Leveraging the Lagrangian formulation of the Fokker-Planck equation, also known as the *probability flow* formulation [15], PFI infers the drift of an Itô diffusion interpolating between the distributions at successive times. A limitation of PFI is that it does not account for growth (i.e. proliferation and death), a key feature of realistic biological systems that must be appropriately modelled for the inference of accurate trajectories. This was notably observed in reprogramming experiments, where neglecting cellular death led to the inference of incorrect state transitions between apoptotic and pluripotent cells [2]. In this paper, we show how the PFI approach naturally extends to account for cellular proliferation, and we propose a new algorithm called *unbalanced probability flow inference* (UPFI), which allows to disentangle drift from intrinsic noise and growth in diffusion processes.

**Stochastic population dynamics with growth**   We consider a population of individuals evolving following an Itô process of the form

$$d\boldsymbol{X}_t = \boldsymbol{v}_t(\boldsymbol{X}_t)\,dt + \boldsymbol{\sigma}_t(\boldsymbol{X}_t)d\boldsymbol{B}_t, \tag{1}$$

with $d\boldsymbol{B}_t$ the increments of a $d$-dimensional Brownian motion, $\boldsymbol{v}_t(\boldsymbol{X}_t)$ the drift encoding deterministic aspects of the system and $\boldsymbol{\sigma}_t(\boldsymbol{X}_t)$ the strength of the noise, which can depend on the state $\boldsymbol{X}_t$, as well as the drift.

To include growth in the model (1), we model individuals as undergoing *division* and *death* at rates $b_t(\boldsymbol{X}_t)$ and $d_t(\boldsymbol{X}_t)$, respectively. More precisely, during a short time interval $\tau \ll 1$, an individual with state $\boldsymbol{X}_t$ divides with probability $b_t(\boldsymbol{X}_t)\tau + o(\tau)$ and dies with probability $d_t(\boldsymbol{X}_t)\tau + o(\tau)$. Upon a division event, both descendants inherit their parent's state $\boldsymbol{X}_t$. With these prescriptions, the Itô process (1), along with the division and death mechanism, defines a branching diffusion process [16]. The density of individuals with state $\boldsymbol{x}$ at time $t$, denoted $\rho_t(\boldsymbol{x})$, follows a Fokker-Planck equation (FPE) with a source term [12]:

$$\partial_t \rho_t(\boldsymbol{x}) = -\boldsymbol{\nabla} \cdot [\rho_t(\boldsymbol{x}) v_t(\boldsymbol{x}) - \boldsymbol{\nabla} \cdot (\rho_t(\boldsymbol{x})\boldsymbol{D}_t(\boldsymbol{x}))] + g_t(\boldsymbol{x})\rho_t(\boldsymbol{x}), \tag{2}$$

where $\boldsymbol{D}_t(\boldsymbol{x}) = \frac{1}{2}\boldsymbol{\sigma}_t(\boldsymbol{x})\boldsymbol{\sigma}_t(\boldsymbol{x})^\top \in \mathbb{S}_+^d$ is the diffusivity matrix and $g_t(\boldsymbol{x}) = b_t(\boldsymbol{x}) - d_t(\boldsymbol{x})$ is the net fitness associated with the gene expression $\boldsymbol{x}$. Regions where $g_t(\boldsymbol{x}) < 0$ correspond to regions of net death, while regions $g_t(\boldsymbol{x}) > 0$ correspond to net growth. The total number of cells $|\rho_t| = \int \rho_t(\boldsymbol{x})d\boldsymbol{x}$ is thus not conserved over time, and $\rho_t$ is not a probability density. In Fig. 1(i) we illustrate the evolution of $\rho_t$ for a bistable process in two dimensions, where growth has the effect of accumulating the mass on one of the two branches, even though both branches are equally favourable in terms of the energetic landscape. In Section 3, we will analyse in more detail a high-dimensional version of this bifurcating process.

To motivate the relevance of (1) for cell dynamics, we point out that a SDE of this form with coupled drift and noise terms naturally arises from the Chemical Langevin Equation, obtained by coarse-graining of the biophysically principled Chemical Master Equation [17]. The importance of multiplicative noise in biological systems is also pointed out by [3] which highlights the nontrivial impact that complex noise models can have on dynamics. Finally, multiplicative noise models also arise in non-biological settings such as the Cox-Ingersoll-Ross model in finance [18].

On the other hand, the importance of modelling division and death is readily apparent for applications such as cell dynamics, ecology, and disease modelling [19, 20]. In biological systems, organised cell division and death are a key control mechanism that work alongside regulatory dynamics, and are thus an essential component that must be modelled to ensure accurate inference results.

**Problem statement**  Population snapshot measurements arise in settings where tracking of individuals is either impossible or expensive, such as in single-cell RNA sequencing (scRNA-seq) studies. Longitudinal information on the stochastic trajectories of individuals and their progenitors is lost. Instead, one has access only to *statistically independent*, cross-sectional measurements of the population density $\rho_t(\boldsymbol{x})$. In other words, we observe the *temporal marginals* of the process (1) with birth and death. Given the knowledge of $\rho_t(\boldsymbol{x})$, our goal is to infer the drift $\boldsymbol{v}_t$, the fitness $g_t$ and the diffusivity $\boldsymbol{D}_t(\boldsymbol{x})$ of the underlying branching diffusion process, or in other words, to invert (2). While our approach applies to the simultaneous inference of these three quantities, we assume that $\boldsymbol{D}_t(\boldsymbol{x})$, or the form of its functional dependency on the drift in the case of a Chemical Langevin Equation [17], is known. In what follows, we restrict ourselves to inferring the drift and the fitness.

Consider $K + 1$ cross-sectional measurements taken from (2) at successive times $t_0 = 0 < \cdots < t_K = T$. Each measurement $i$ consists of $N_i$ sampled states $\{\boldsymbol{x}_{k,t_i}, 1 \leqslant k \leqslant N_i\}$. We assume that $N_i/N_0$ estimates the ratio of individuals $|\rho_{t_i}|/|\rho_0|$ at all times $t_i$, $1 \leqslant i \leqslant K$. With this assumption, we can estimate $\rho_{t_i}$ up to a multiplicative constant $|\rho_0|/N_0$ using the samples available: $\rho_{t_i} \simeq \frac{|\rho_0|}{N_0} \sum_{j=1}^{N_i} \delta(\boldsymbol{x} - \boldsymbol{x}_{k,t_i})$. Practically, the knowledge of this constant term is unnecessary because the rescaled density $N_0 \rho_t / |\rho_0|$ satisfies the same equation as $\rho_t$ so we set it to one.

**Related work**  Approaches based on optimal transport for reconstructing dynamics typically either ignore cellular birth and death entirely [11, 13, 14, 21], or handle growth using an unbalanced relaxation of entropic optimal transport (EOT) and prior knowledge on growth [2, 9, 10, 12, 22]. In particular, while unbalanced EOT can easily be solved computationally using a modified Sinkhorn algorithm [23], it proceeds via a relaxation of the static transport problem and does not readily offer a natural dynamical interpretation in the sense of the continuous-time dynamics of (2). Recently a significant amount of theoretical progress was made on the closely related Schrödinger bridge problem for branching Brownian motion [16], however designing computational approaches for its solution in high dimensional settings remains an open problem. Several recent works aim to learn neural approximations to dynamics with growth, but either assume deterministic dynamics [24], a global fitness linking drift to growth [25] (see also the discussion in Appendix C.3), or rely in the end on unbalanced EOT for conditioning flow matching [26]. These methods are therefore subject to limitations rendering them inapplicable for the problem we describe. One work that addresses our problem setting is [27] which proposes a deep learning based method, DeepRUOT. While the algorithm is designed with the same aims as ours, it requires an multi-stage training procedure prone to instability, first relying on flow matching and then on simulation-based training.

## 2  Methodology

**Unbalanced Probability Flow Inference (UPFI)**  To solve this inverse problem we build upon the Probability Flow Inference (PFI) approach of [13, 14], in which the Fokker-Planck equation is fitted in the Lagrangian frame of reference. This formulation leverages the fact that the drift-diffusion term in (2) can be re-written as a transport term. In our setting, such a rewriting reads

$$\partial_t \rho_t(\boldsymbol{x}) = -\boldsymbol{\nabla} \cdot [\rho_t(\boldsymbol{x}) \left(\boldsymbol{v}_t(\boldsymbol{x}) - \boldsymbol{\nabla} \cdot \boldsymbol{D}_t(\boldsymbol{x}) - \boldsymbol{D}_t(\boldsymbol{x}) \boldsymbol{\nabla} \log \rho_t(\boldsymbol{x})\right)] + g_t(\boldsymbol{x}) \rho_t(\boldsymbol{x}), \qquad (3)$$

The phase-space velocity in the transport term now depends on the score $\boldsymbol{\nabla} \log \rho_t(\boldsymbol{x})$, which is independent of the total mass of the measure since $\boldsymbol{\nabla} \log \rho_t = \boldsymbol{\nabla} \log (\rho_t / |\rho_t|)$. Interestingly, (3) can be solved in the same Lagrangian frame of reference as without growth $g_t = 0$. Indeed, the

solution $(\boldsymbol{x}_t, m_t)$ of the following system of ODEs

$$
\begin{aligned}
\frac{\mathrm{d}\boldsymbol{x}_t}{\mathrm{d}t} &= \boldsymbol{v}_t(\boldsymbol{x}_t) - \boldsymbol{\nabla} \cdot \boldsymbol{D}_t(\boldsymbol{x}_t) - \boldsymbol{D}_t(\boldsymbol{x}_t)\boldsymbol{\nabla}\log\rho_t(\boldsymbol{x}_t) =: \boldsymbol{u}_t(\boldsymbol{x}_t), & \boldsymbol{x}_0 &\sim \rho_0, \\
\frac{\mathrm{d}m_t}{\mathrm{d}t} &= g_t(\boldsymbol{x}_t)m_t, & m_0 &= \rho_0(\boldsymbol{x}_0),
\end{aligned}
\tag{4}
$$

produces a weak solution to (2) [28, Proposition 2.1], where we have written $\boldsymbol{u}_t$ to denote the phase-space velocity field. In other terms, by going to the Lagrangian frame of reference, we can trade solving a PDE in $d$ dimensions with solving an ODE system in $d+1$ dimensions. This gain comes at a cost, since it requires learning the score function $\boldsymbol{\nabla}\log\rho_t(\boldsymbol{x})$ beforehand. However, this can efficiently be done offline for high-dimensional datasets because the score is independent of the parameters of the dynamical model in (4).

We remark on the importance that (4) yields a *weak* solution to (2) – solving this system for finitely many particles yields a weighted sum of point masses $\widehat{\rho}_t$ approximating the measure $\rho_t \, \mathrm{d}x$ but does not provide the *density* $\rho_t$. The density evolution is governed by $\mathrm{d}(\log\rho_t(\boldsymbol{x}_t))/\mathrm{d}t = g_t(\boldsymbol{x}_t) - (\nabla \cdot \boldsymbol{u}_t)(\boldsymbol{x}_t)$, i.e. an additional term arises from the divergence of the phase-space velocity. While this is the setting arising in the continuous normalising flows literature [29] and is also used by [24] in the context of growth, in practice computing the divergence term is well-known to be computationally expensive as it relies on computing the trace of a Jacobian.

Our approach, UPFI, consists of two steps: (i) estimating a time-dependent score function $\boldsymbol{s}_t(\boldsymbol{x}) \simeq \boldsymbol{\nabla}\log\rho_t(\boldsymbol{x})$ from available snapshots, and (ii) learning the coefficients of the ODE system (4) that fit the observed snapshots $\{\rho_{t_i}, 1 \leqslant i \leqslant K\}$. For the first step of the algorithm, we estimate the score with denoising score matching [5, 30], as it has shown best performance in terms of accuracy and scalability on similar problems [14]. For the second step, we push the observed samples and their associated mass from time $t_i$ to time $t_{i+1}$ following (4). The explicit update equations read

$$
\widehat{\boldsymbol{x}}_{k,t} = \boldsymbol{x}_{k,t_i} + \int_{t_i}^{t} \left(\boldsymbol{v}_u(\widehat{\boldsymbol{x}}_{k,u}) - \boldsymbol{\nabla} \cdot \boldsymbol{D}_u(\widehat{\boldsymbol{x}}_{k,u}) - \boldsymbol{D}_u(\widehat{\boldsymbol{x}}_{k,u})\boldsymbol{s}_u(\widehat{\boldsymbol{x}}_{k,u})\right)\mathrm{d}u \tag{5}
$$

$$
m_{k,t} = \exp\left(\int_{t_i}^{t} g_u(\widehat{\boldsymbol{x}}_{k,u})\mathrm{d}u\right), \tag{6}
$$

with $t \in [t_i, t_{i+1}]$. This allows us to construct an inferred marginal distribution $\widehat{\rho}_{t_{i+1}}$, which reads

$$
\widehat{\rho}_{t_{i+1}}(\boldsymbol{x}) = \sum_{k=1}^{N_i} m_{k,t_{i+1}}\delta(\widehat{\boldsymbol{x}}_{k,t_{i+1}} - \boldsymbol{x}). \tag{7}
$$

We can then optimise jointly over the drift and the growth by minimising a discrepancy $\mathcal{D}(\widehat{\rho}_{t_{i+1}}, \rho_{t_{i+1}})$ between the inferred and the true measures. In practice, we minimise the total discrepancy across all snapshots, $(\boldsymbol{v}_t^\star(\boldsymbol{x}), g_t^\star(\boldsymbol{x})) = \arg\min_{(\boldsymbol{v}_t(\boldsymbol{x}), g_t(\boldsymbol{x}))} \sum_{i=1}^{K} \mathcal{D}(\widehat{\rho}_{t_i}, \rho_{t_i})$. While in principle various choices of $\mathcal{D}$ could be appropriate, we opt to use the unbalanced Sinkhorn divergence [31] which possesses desirable geometric and computational properties [23, 32]: $\mathcal{D} = \mathrm{S}_{\varepsilon,\gamma}$, where $\varepsilon, \gamma > 0$ specify the entropic regularisation level and soft mass constraint respectively. Most importantly, this choice works directly with discrete measures, allowing us to bypass entirely computation of densities.

We summarise the UPFI algorithm in Alg. 1. These successive steps are illustrated for a two-dimensional bifurcating process in Fig. 1(ii)-(iii). In Fig. 1(iv) we show the difference between the resulting probability flow lines inferred with PFI and with UPFI. Because PFI doesn't explicitly account for proliferation, it infers flow lines which correct for the mass imbalance between the two bifurcating branches by incorrectly connecting them. On the other hand, UPFI recovers the appropriate flow lines which do not connect the two bifurcating branches. The inference of a drift biased by proliferation, as exemplified with PFI in Fig.1D, is an instance of a wider issue regarding the ability to infer jointly drift of proliferation.

**The problem of identifying simultaneously drift and fitness** Even in the absence of cellular proliferation, the drift term $\boldsymbol{v}_t(\boldsymbol{x})$ is in general not uniquely identifiable [14, 33, 34]. Naturally, these same arguments extend to the simultaneous inference of the drift and growth, and furthermore we can expect to mistake drift for growth and vice versa. While the approach outlined in Algorithm 1 applies to any nonlinear force field, here we aim to gain theoretical insight into the identifiability problem by analytically studying an Ornstein-Uhlenbeck (OU) process with quadratic fitness, in which the population density remains a Gaussian measure. Although OU processes can not capture bifurcations, they have been successfully used to infer non-bifurcating processes in single-cell RNA-seq data [35]. For these processes, the drift is a linear function $\boldsymbol{v}_t(\boldsymbol{x}) = \boldsymbol{A}_t\boldsymbol{x} + \boldsymbol{e}_t$ and the matrix

**Algorithm 1** Unbalanced Probability Flow Inference (UPFI)

---

**Input:** $K + 1$ statistically independent snapshots $\{\boldsymbol{x}_{k,t_i}\}_{k=1}^{N_i}$ sampled from $\{\rho_{t_i}\}_{i=0}^{K}$, i.e. the solution to (2) at successive times $t_0, t_1, \ldots t_K$.
**Estimate score:** $\boldsymbol{s}_\phi(t, \boldsymbol{x}) \approx \boldsymbol{\nabla} \log \rho_t(\boldsymbol{x})$ using score matching
**Initialise:** Force $\boldsymbol{v}_\theta(t, \boldsymbol{x})$, growth rate $\boldsymbol{g}_\theta(t, \boldsymbol{x})$, diffusivity $\boldsymbol{D}(t, \boldsymbol{x})$.
**while** not converged **do**
    **for** $k \in \{0, 1, \ldots, K - 1\}$ **do**
        $(\boldsymbol{x}_{\ell,t_{k+1}}, m_{\ell,t_{k+1}})_{\ell=1}^{N_k} \leftarrow \texttt{odesolve}(\text{Eq. (4)}, \texttt{x0} = (\boldsymbol{x}_{\ell,t_k}, m_{\ell,t_k})_{\ell=1}^{N_k}, \texttt{ts} = [t_k, t_{k+1}])$
        $\widehat{\rho}_{t_{k+1}} \leftarrow \sum_{\ell=1}^{N_k} m_{\ell,t_{k+1}} \delta(\boldsymbol{x} - \boldsymbol{x}_{\ell,t_{k+1}})$           *Predicted marginal at $t_{k+1}$*
        $\ell_k \leftarrow \mathrm{S}_{\varepsilon,\gamma}(\rho_{t_{k+1}}, \widehat{\rho}_{t_{k+1}})$           *Sinkhorn loss on marginals*
    $\mathcal{L} = K^{-1} \sum_{k=1}^{K-1} \ell_k$           *Compute total loss*
    $\theta \leftarrow \mathrm{Update}(\theta, \boldsymbol{\nabla}_\theta \mathcal{L})$           *Update $\theta$*
**return** $\boldsymbol{v}_\theta$

---

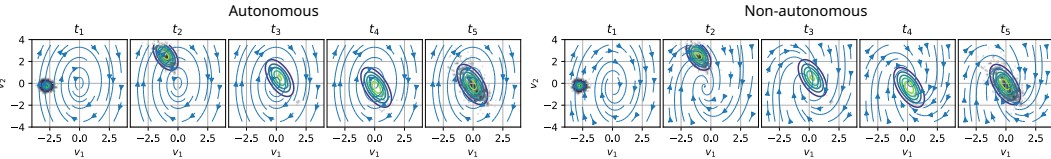

Figure 2: **Non-identifiability of growth and drift in the Gaussian case.** 8-dimensional OU process with quadratic growth. Left: autonomous drift and growth produces the same marginals as a non-autonomous system.

$\boldsymbol{A}_t$ is directly interpretable as the gene regulatory network driving the process. We prove in the appendix the following result for OU processes with quadratic fitness.

**Proposition 2.1** (OU process with quadratic fitness). *Consider an OU process with quadratic fitness, whose density satisfies*

$$\partial_t \rho_t(\boldsymbol{x}) = -\boldsymbol{\nabla} \cdot ((\boldsymbol{A}_t \boldsymbol{x} + \boldsymbol{e}_t)\rho_t(\boldsymbol{x}) - \boldsymbol{D}_t \boldsymbol{\nabla}\rho_t(\boldsymbol{x})) + (b_t + \boldsymbol{c}_t^\top \boldsymbol{x} + \frac{1}{2}\boldsymbol{x}^\top \boldsymbol{\Gamma}_t \boldsymbol{x})\rho_t(\boldsymbol{x}), \quad (8)$$

*where $\boldsymbol{A}_t \in \mathbb{R}^{d \times d}$, $\boldsymbol{e}_t, \boldsymbol{c}_t \in \mathbb{R}^d$, and $b_t \in \mathbb{R}$ are generic, $\boldsymbol{D}_t \in \mathbb{R}^{d \times d}$ is symmetric positive definite, and $\boldsymbol{\Gamma}_t \in \mathbb{R}^{d \times d}$ is symmetric negative semi-definite. If $\rho_0 = m_0 \mathcal{N}(\boldsymbol{\mu}_0, \boldsymbol{\Sigma}_0)$ with $m_0 > 0$, then $\rho_t = m_t \mathcal{N}(\boldsymbol{\mu}_t, \boldsymbol{\Sigma}_t)$ for all $t \geqslant 0$, where*

$$\dot{\boldsymbol{\Sigma}}_t = \boldsymbol{A}_t \boldsymbol{\Sigma}_t + \boldsymbol{\Sigma}_t \boldsymbol{A}_t^\top + 2\boldsymbol{D}_t + \boldsymbol{\Sigma}_t \boldsymbol{\Gamma}_t \boldsymbol{\Sigma}_t, \quad (9)$$

$$\dot{\boldsymbol{\mu}}_t = (\boldsymbol{A}_t + \boldsymbol{\Sigma}_t \boldsymbol{\Gamma}_t)\boldsymbol{\mu}_t + \boldsymbol{\Sigma}_t \boldsymbol{c}_t + \boldsymbol{e}_t, \quad (10)$$

$$\frac{\dot{m_t}}{m_t} = \frac{1}{2}\boldsymbol{\mu}_t^\top \boldsymbol{\Gamma}_t \boldsymbol{\mu}_t + \boldsymbol{c}_t^\top \boldsymbol{\mu}_t + b_t + \frac{1}{2}\mathrm{tr}(\boldsymbol{\Gamma}_t \boldsymbol{\Sigma}_t). \quad (11)$$

Using this result, we can observe that it is impossible to identify the gene regulatory network uniquely, and that many solutions fit equally well the data. This is made concrete in the following result.

**Corollary 2.2.** *Consider an OU process with drift $\boldsymbol{v}_t(\boldsymbol{x}) = \boldsymbol{A}_t \boldsymbol{x}$ and a time-dependent fitness as in (8). Let $\boldsymbol{K}_t \in \mathbb{R}^{n \times n}$ be an arbitrary matrix. Then there exists $\epsilon > 0$ such that the system is indistinguishable from another OU process with drift $\boldsymbol{v}_t(\boldsymbol{x}) = \boldsymbol{A}_t + \boldsymbol{I} + \epsilon\boldsymbol{K}_t$ and time-dependent quadratic fitness.*

Importantly, this result holds even if we enforce the drift to be autonomous: both symmetric and asymmetric parts of the drift matrix can be confounded with cellular growth, and vice-versa. We show in Fig. 2 a simple illustration of this fact – the same sequence of distributions can arise from autonomous linear drift with quadratic growth, but also non-autonomous linear drift alone.

**Loss function and uniqueness of the minimum** The identifiability issue discussed previously calls for the use of a regularisation when optimising for the drift $\boldsymbol{v}_t(\boldsymbol{x})$. This regularisation ensures that, even if the method is not consistent in general, it leads at least to a unique solution. While different regularisers could be used, here we opt for the Wasserstein-Fisher-Rao energy function

[32] as a natural choice. The loss function reads, with $\alpha > 0, \lambda \geqslant 0$:

$$L = \sum_{i=1}^{K} S_{\varepsilon,\gamma}(\widehat{\rho}_{t_i}, \rho_{t_i}) + \lambda(t_i - t_{i-1}) \int_{t_{i-1}}^{t_i} \int \left( \|\boldsymbol{v}_t(\boldsymbol{x})\|^2 + \alpha |g_t(\boldsymbol{x})|^2 \right) \, \mathrm{d}\rho_t(\boldsymbol{x}) \, \mathrm{d}t. \quad (12)$$

Once again, we can gain some insight into the role of the regularisation by studying the linear-quadratic case from Prop. 2.1. Assuming $t_i - t_{i-1} = \Delta t$, we show in the appendix that in the non-entropic limit $\varepsilon \to 0$ (in which case the Sinkhorn divergence becomes the Gaussian Hellinger-Kantorovich distance [36]) and when $\Delta t \to 0$ with $T = K\Delta t$ fixed, the loss tends to a functional which has a unique minimum as a function of the parameters defining the fitness and the drift.

**Theorem 2.3** (Loss function for OU processes with quadratic fitness). *Consider the true process as well as the inferred processed to both be OU processes with quadratic fitness, i.e.* (8). *Denoting* $q = 2/\gamma$, *with* $K\Delta t$ *fixed, when* $\Delta t \to 0$ *we have* $\Delta t^{-1} L \to \mathcal{L}$, *where* $\mathcal{L}$ *is the continuous time loss function:*

$$\mathcal{L} = q^{-1} \int_0^T \mathrm{d}t \, m_t \bigg( \|\boldsymbol{v}_t - \widehat{\boldsymbol{v}}_t\|_{\boldsymbol{X}_t^{-1}}^2 + \frac{1}{2}(h_t - \widehat{h}_t)^2 \qquad\qquad\qquad (13)$$

$$+ q \sum_{i,j} \frac{\sigma_{i,t}^2 \sigma_{j,q,t}^2}{(\sigma_{j,q,t}^2 \sigma_{i,t}^2 + \sigma_{i,q,t}^2 \sigma_{j,t}^2)^2} \left( \boldsymbol{w}_{i,t}^\top (\boldsymbol{B}_t - \widehat{\boldsymbol{B}}_t) \boldsymbol{w}_{j,t} \right)^2 \bigg) + \lambda \int_0^T R_t \, \mathrm{d}t$$

$$(14)$$

*where* $\boldsymbol{\Sigma}_t = \sum_i \sigma_{i,t}^2 \boldsymbol{w}_{i,t} \boldsymbol{w}_{i,t}^\top$ *is the eigendecomposition of the covariance* $\boldsymbol{\Sigma}_t$, $\sigma_{i,q,t}^2 = 1 + q\sigma_{i,t}^2$ *for all* $i$, $\boldsymbol{X}_t = 2\boldsymbol{\Sigma}_t + q^{-1}$. *We also have*

$$\boldsymbol{B}_t - \widehat{\boldsymbol{B}}_t = \boldsymbol{\Sigma}_t(\boldsymbol{A}_t - \widehat{\boldsymbol{A}}_t)^\top + (\boldsymbol{A}_t - \widehat{\boldsymbol{A}}_t)\boldsymbol{\Sigma}_t + \boldsymbol{\Sigma}_t(\boldsymbol{\Gamma}_t - \widehat{\boldsymbol{\Gamma}}_t)\boldsymbol{\Sigma}_t + 2(\boldsymbol{D}_t - \widehat{\boldsymbol{D}}_t),$$

$$\boldsymbol{v}_t - \widehat{\boldsymbol{v}}_t = ((\boldsymbol{A}_t - \widehat{\boldsymbol{A}}_t) + \boldsymbol{\Sigma}_t(\boldsymbol{\Gamma}_t - \widehat{\boldsymbol{\Gamma}}_t))\boldsymbol{\mu}_t + \boldsymbol{\Sigma}_t(\boldsymbol{c}_t - \widehat{\boldsymbol{c}}_t) + (\boldsymbol{e}_t - \widehat{\boldsymbol{e}}_t), \quad (15)$$

$$h_t - \widehat{h}_t = \frac{1}{2}\boldsymbol{\mu}_t^\top(\boldsymbol{\Gamma}_t - \widehat{\boldsymbol{\Gamma}}_t)\boldsymbol{\mu}_t + (\boldsymbol{c}_t - \widehat{\boldsymbol{c}}_t)^\top \boldsymbol{\mu}_t + (b_t - \widehat{b}_t) + \frac{1}{2}\mathrm{tr}((\boldsymbol{\Gamma}_t - \widehat{\boldsymbol{\Gamma}}_t)\boldsymbol{\Sigma}_t),$$

*and* $R_t$ *is a strongly convex function of the parameters defining the fitness and the drift. For* $\lambda > 0$, $\mathcal{L}$ *has a unique minimiser* $t \mapsto (\widehat{\boldsymbol{A}}_t^\star, \widehat{\boldsymbol{e}}_t^\star, \widehat{b}_t^\star, \widehat{\boldsymbol{c}}_t^\star, \widehat{\boldsymbol{\Gamma}}_t^\star)$.

**Scalability** The unbalanced Sinkhorn divergence $S_{\varepsilon,\gamma}$ can be computed with a per-iteration complexity of $\mathcal{O}(B^2)$ and a dimension-independent sample complexity of $\mathcal{O}(B^{-1/2})$ [31], provided the batch size $B$ is sufficiently large. In contrast, denoising score matching has a per-iteration complexity of $\mathcal{O}(Bd)$, where $d$ denotes the dimensionality. The computational cost of ODE integration can be reduced, at the expense of some accuracy, by taking fewer Euler steps between time points. In practice, we find that two to three steps are typically sufficient. This imposes a mild limitation on the scalability to high dimensions, and we find the UPFI algorithm can handle up to moderately high dimensions as illustrated in the results section below. This moderately high-dimensional regime is particularly relevant for modelling cellular processes such as haematopoietic stem cell differentiation, where only a few dozen key transcription factors drive cell fate decisions [1]. Additionally, PCA dimensionality reduction is a routine preprocessing step in single cell analysis workflows [37].

## 3 Results

**High-dimensional bifurcating system** We first consider a bistable system in $\mathbb{R}^d$ where the dynamics are driven by a potential landscape $\boldsymbol{v} = -\nabla V$, $V(\boldsymbol{x}) = 0.9\|\boldsymbol{x} - \boldsymbol{a}\|_2^2 \|\boldsymbol{x} - \boldsymbol{b}\|_2^2 + 10 \sum_{i=3}^d x_i^2$, where attractors are located at $\boldsymbol{a}, \boldsymbol{b} = \pm(\boldsymbol{e}_1 + \boldsymbol{e}_2)$. We impose a birth rate $b(\boldsymbol{x}) = \frac{5}{2}(1 + \tanh(2x_0))$ and $d(\boldsymbol{x}) = 0$ so that individuals closer to the positive attractor divide at a high rate (see schematic in Fig. 1(i)). For $d = \{2, 5, 10, 25, 50\}$ we simulate the system and sample population snapshots at $T = 5$ timepoints (see Fig. 3(a) for $d = 10$). To reconstruct the dynamics, we apply UPFI (Alg. 1), as well as PFI [13, 14]. For additional comparisons, we consider a principled deterministic fitness-based method (fitness-ODE, see Section C.3), a method inspired by TIGON [24], DeepRUOT [27], optimal transport flow matching (OTFM) [38] and its unbalanced variant, UOTFM [26]. The implementation of TIGON as described and provided by [24] is difficult to use in practice, we therefore re-implemented the method with several simplifications which we believe improves its performance,

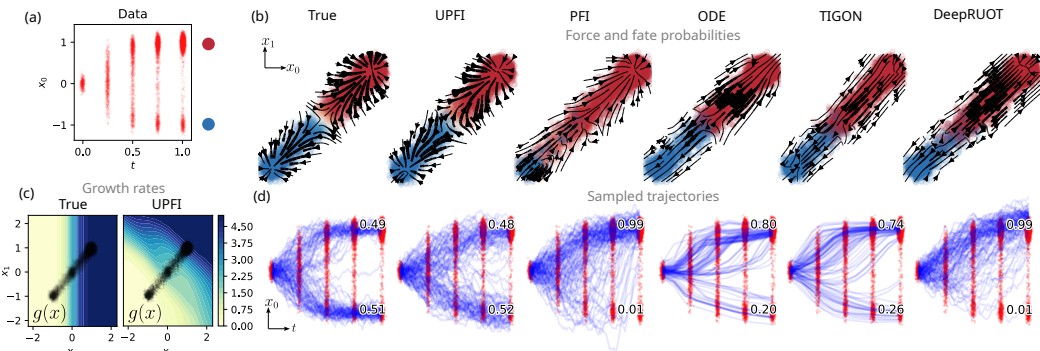

Figure 3: **10-dimensional bistable system.** (a) Population snapshots over time, shown in the first coordinate $x_0$. (b) True and inferred force fields shown in $(x_0, x_1)$, coloured by fate probabilities. (c) True and inferred growth rates shown in $(x_0, x_1)$. (d) Sampled trajectories from true and learned dynamics with growth suppressed (pure drift-diffusion process). The fraction of trajectories terminating in the upper (resp. lower) regions are indicated. Without growth both branches are equiprobable.

| $d$ | UPFI | PFI | FITNESS-ODE | TIGON++ | DEEPRUOT | OTFM | UOTFM |
|---|---|---|---|---|---|---|---|
| | | | **Path energy distance ($\downarrow$)** | | | | |
| 2 | **0.14 $\pm$ 0.09** | 1.41 $\pm$ 0.16 | 0.30 $\pm$ 0.18 | 0.46 $\pm$ 0.12 | 2.15 $\pm$ 0.01 | 1.16 $\pm$ 0.13 | 0.42 $\pm$ 0.13 |
| 5 | **0.04 $\pm$ 0.03** | 1.34 $\pm$ 0.06 | 0.30 $\pm$ 0.14 | 0.63 $\pm$ 0.16 | 0.47 $\pm$ 0.04 | 1.07 $\pm$ 0.11 | 0.36 $\pm$ 0.10 |
| 10 | **0.05 $\pm$ 0.04** | 1.03 $\pm$ 0.18 | 0.29 $\pm$ 0.15 | 0.61 $\pm$ 0.06 | 1.32 $\pm$ 0.05 | 1.09 $\pm$ 0.19 | 0.38 $\pm$ 0.08 |
| 25 | 0.22 $\pm$ 0.20 | 1.51 $\pm$ 0.10 | 0.20 $\pm$ 0.04 | 0.74 $\pm$ 0.04 | **0.14 $\pm$ 0.00** | 1.39 $\pm$ 0.09 | 0.57 $\pm$ 0.22 |
| 50 | **0.15 $\pm$ 0.02** | 1.89 $\pm$ 0.21 | 0.20 $\pm$ 0.08 | 0.62 $\pm$ 0.04 | 0.31 $\pm$ 0.02 | 1.38 $\pm$ 0.06 | 0.69 $\pm$ 0.14 |
| | | | **Force error (cosine $\downarrow$)** | | | | |
| 2 | **0.07 $\pm$ 0.02** | 0.08 $\pm$ 0.01 | 0.34 $\pm$ 0.04 | 0.26 $\pm$ 0.04 | 0.44 $\pm$ 0.00 | 0.41 $\pm$ 0.03 | 0.44 $\pm$ 0.06 |
| 5 | 0.15 $\pm$ 0.04 | **0.09 $\pm$ 0.01** | 0.36 $\pm$ 0.02 | 0.37 $\pm$ 0.03 | 0.44 $\pm$ 0.00 | 0.44 $\pm$ 0.01 | 0.46 $\pm$ 0.03 |
| 10 | **0.10 $\pm$ 0.00** | 0.12 $\pm$ 0.00 | 0.37 $\pm$ 0.05 | 0.35 $\pm$ 0.02 | 0.45 $\pm$ 0.00 | 0.45 $\pm$ 0.01 | 0.45 $\pm$ 0.01 |
| 25 | **0.06 $\pm$ 0.00** | 0.09 $\pm$ 0.00 | 0.19 $\pm$ 0.04 | 0.27 $\pm$ 0.01 | 0.37 $\pm$ 0.00 | 0.60 $\pm$ 0.01 | 0.63 $\pm$ 0.02 |
| 50 | **0.06 $\pm$ 0.00** | 0.07 $\pm$ 0.00 | 0.26 $\pm$ 0.03 | 0.25 $\pm$ 0.01 | 0.44 $\pm$ 0.00 | 0.55 $\pm$ 0.01 | 0.52 $\pm$ 0.01 |
| | | | **Force error ($L^2$, $\downarrow$)** | | | | |
| 2 | **1.17 $\pm$ 0.03** | 1.81 $\pm$ 0.09 | 1.70 $\pm$ 0.04 | 1.62 $\pm$ 0.03 | 2.84 $\pm$ 0.00 | 1.77 $\pm$ 0.06 | 1.68 $\pm$ 0.04 |
| 5 | **2.89 $\pm$ 0.11** | 3.53 $\pm$ 0.18 | 3.78 $\pm$ 0.02 | 3.83 $\pm$ 0.04 | 4.07 $\pm$ 0.00 | 3.86 $\pm$ 0.01 | 3.80 $\pm$ 0.03 |
| 10 | **4.31 $\pm$ 0.08** | 4.76 $\pm$ 0.14 | 5.87 $\pm$ 0.13 | 5.89 $\pm$ 0.04 | 6.17 $\pm$ 0.00 | 6.09 $\pm$ 0.04 | 5.96 $\pm$ 0.01 |
| 25 | **6.74 $\pm$ 0.05** | 6.95 $\pm$ 0.05 | 8.94 $\pm$ 0.20 | 9.58 $\pm$ 0.03 | 9.98 $\pm$ 0.00 | 10.32 $\pm$ 0.02 | 10.28 $\pm$ 0.04 |
| 50 | 9.45 $\pm$ 0.06 | **9.41 $\pm$ 0.03** | 13.37 $\pm$ 0.22 | 13.43 $\pm$ 0.04 | 14.45 $\pm$ 0.00 | 14.60 $\pm$ 0.04 | 14.49 $\pm$ 0.01 |
| | | | **Pearson fate correlation, $\uparrow$** | | | | |
| 2 | **0.98 $\pm$ 0.01** | 0.57 $\pm$ 0.04 | 0.94 $\pm$ 0.01 | 0.91 $\pm$ 0.03 | 0.51 $\pm$ 0.00 | 0.79 $\pm$ 0.01 | 0.96 $\pm$ 0.01 |
| 5 | **0.99 $\pm$ 0.00** | 0.59 $\pm$ 0.02 | 0.93 $\pm$ 0.01 | 0.82 $\pm$ 0.04 | 0.84 $\pm$ 0.00 | 0.76 $\pm$ 0.01 | 0.95 $\pm$ 0.00 |
| 10 | **0.99 $\pm$ 0.00** | 0.65 $\pm$ 0.02 | 0.93 $\pm$ 0.01 | 0.84 $\pm$ 0.03 | 0.77 $\pm$ 0.00 | 0.76 $\pm$ 0.02 | 0.95 $\pm$ 0.00 |
| 25 | **0.97 $\pm$ 0.01** | 0.65 $\pm$ 0.02 | 0.90 $\pm$ 0.02 | 0.77 $\pm$ 0.01 | 0.95 $\pm$ 0.00 | 0.74 $\pm$ 0.02 | 0.93 $\pm$ 0.01 |
| 50 | **0.98 $\pm$ 0.00** | 0.63 $\pm$ 0.03 | 0.91 $\pm$ 0.01 | 0.80 $\pm$ 0.02 | 0.92 $\pm$ 0.00 | 0.73 $\pm$ 0.01 | 0.93 $\pm$ 0.01 |

Table 1: Numerical evaluation results for $d$-dimensional bistable system, $d \in \{2, 5, 10, 25, 50\}$.

and we thus refer to it as TIGON++ (see Section C.4 for in-depth discussion). We provide details on flow matching in Section C.6.

Fig. 3(b) shows in the $(x_0, x_1)$ plane: (i) the vector field $\boldsymbol{v}_t$ as a streamplot, and (ii) the *fate probabilities* by colour, obtained from the true and reconstructed dynamics. We define the fate probability of a state $(t_i, \boldsymbol{x})$ towards an attractor $\mathcal{A}$ as the frequency of sampled trajectories starting from $(t_i, \boldsymbol{x})$ that terminate closest to that attractor. (see Section C for full experiment details)

We observe that UPFI recovers vector fields and fate probabilities that almost perfectly resemble the ground truth. On the other hand PFI, unable to account for the presence of growth, misattributes the shift in distribution to a spurious drift towards the faster dividing branch. On the other hand, fitness-ODE and TIGON++ are able to pinpoint the bifurcation but produce binary fate probabilities since they are deterministic. Furthermore, the inferred vector fields are qualitatively different from the ground truth for the same reason. DeepRUOT, which in theory models both growth and stochasticity, is unable to recover good dynamics. This is potentially owing to instabilities in their training scheme,

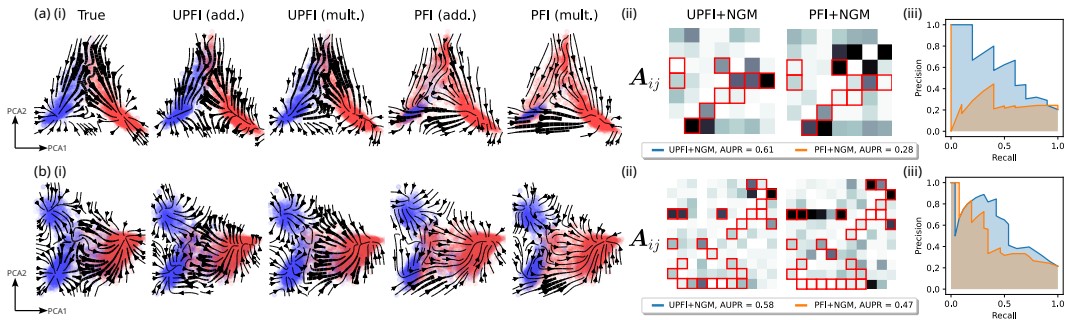

Figure 4: **Simulated regulatory networks.** (a) (i) True and inferred vector fields for 7-dimensional bifurcating system, coloured by fate probabilities. (ii) Learned causal graphs using neural graphical model within UPFI (PFI) frameworks with true interactions shown in red. (iii) Precision-recall curve quantification of prediction accuracy. (b)(i-iii) Same as (a) for 11-dimensional HSC system.

| | UPFI (add.) | UPFI (mult.) | PFI (add.) | PFI (mult.) |
|---|---|---|---|---|
| | | **Bifurcating** | | |
| Path energy distance ($\downarrow$) | $\mathbf{1.22 \pm 0.38}$ | $1.69 \pm 0.14$ | $1.91 \pm 1.04$ | $4.62 \pm 1.14$ |
| Force error ($L^2$, $\downarrow$) | $12.59 \pm 0.39$ | $\mathbf{11.78 \pm 0.30}$ | $18.96 \pm 3.25$ | $16.49 \pm 1.62$ |
| Pearson fate correlation ($\uparrow$) | $0.97 \pm 0.00$ | $\mathbf{0.98 \pm 0.00}$ | $0.66 \pm 0.17$ | $0.62 \pm 0.06$ |
| | | **Haematopoietic** | | |
| Path energy distance ($\downarrow$) | $0.99 \pm 0.26$ | $\mathbf{0.98 \pm 0.15}$ | $1.42 \pm 0.42$ | $1.86 \pm 0.30$ |
| Force error ($L^2$, $\downarrow$) | $19.58 \pm 0.33$ | $\mathbf{18.13 \pm 0.25}$ | $26.80 \pm 3.71$ | $26.84 \pm 1.48$ |
| Fate prediction error (TV $\downarrow$) | $0.08 \pm 0.00$ | $\mathbf{0.07 \pm 0.00}$ | $0.23 \pm 0.03$ | $0.27 \pm 0.04$ |

Table 2: Numerical evaluation results for CLE systems.

which involves multiple distinct stages and losses [27, Algorithm 1] compared to the two-step UPFI approach (Alg. 1). Visualising the true and inferred growth rates (Fig. 3(c)) we find good agreement. Finally in Fig. 3(d) we show sampled trajectories over time for the true system and each method, and these further illustrate our conclusions from before. Because of space constraints, we show sampled trajectories for OTFM and UOTFM in Fig. 6 in the Appendix.

In Table 1 we show numerical evaluation metrics for each method as $d$ varies; here we also compare to results for OTFM and UOTFM. At the level of trajectories, we quantify the *energy distance* [39] between sampled paths from the ground truth and inferred processes treated as empirical distributions in $L^2([0,1], \mathbb{R}^d)$. At the level of the force, we measure the reconstruction error on $v_t$ in terms of the $L^2$ and cosine distances. We also report the Pearson correlation between the true and estimated fate probabilities. In all cases, we find that UPFI recovers dynamics with high accuracy, while remaining competitive in terms of runtime, as show in Table 7. Finally, in Table 8 we also present regularisation ablation experiments which show limited effect of the growth regularisation term $\alpha$. We interpret this as the evidence of implicit regularisation stemming from relatively small neural network sizes.

**Simulated gene regulatory networks** Next we study more complex systems where dynamics arise from chemical reaction networks via the Chemical Langevin Equation (CLE) [14, 40]: we consider a 7-gene bifurcating system (Fig. 4(a)) and a 11-gene haematopoietic stem cell (HSC) system (Fig. 4(b)) in which the simulated dynamics reflect known biology [41]. Cells in the bifurcating system branch and randomly proceed to one of two stable states. The HSC system is multifurcating and cell states evolve towards one of four stable states, {Monocyte (Mo), Granulocyte (G), Megakaryocyte (Mk), Erythrocyte (E)}. Dynamics in both sys-

| | UPFI | UPFI (Jac) | PFI | PFI (Jac) |
|---|---|---|---|---|
| | | **Bifurcating** | | |
| AUPR ($\uparrow$) | $\mathbf{0.64 \pm 0.03}$ | $0.52 \pm 0.07$ | $0.33 \pm 0.09$ | $0.34 \pm 0.10$ |
| AUROC ($\uparrow$) | $\mathbf{0.79 \pm 0.02}$ | $0.74 \pm 0.03$ | $0.65 \pm 0.07$ | $0.63 \pm 0.08$ |
| | | **Haematopoietic** | | |
| AUPR ($\uparrow$) | $\mathbf{0.59 \pm 0.04}$ | $0.58 \pm 0.02$ | $0.53 \pm 0.05$ | $0.51 \pm 0.04$ |
| AUROC ($\uparrow$) | $\mathbf{0.80 \pm 0.02}$ | $0.79 \pm 0.01$ | $0.73 \pm 0.02$ | $0.70 \pm 0.02$ |

Table 3: Accuracy of directed graph prediction using neural graphical model for CLE systems.

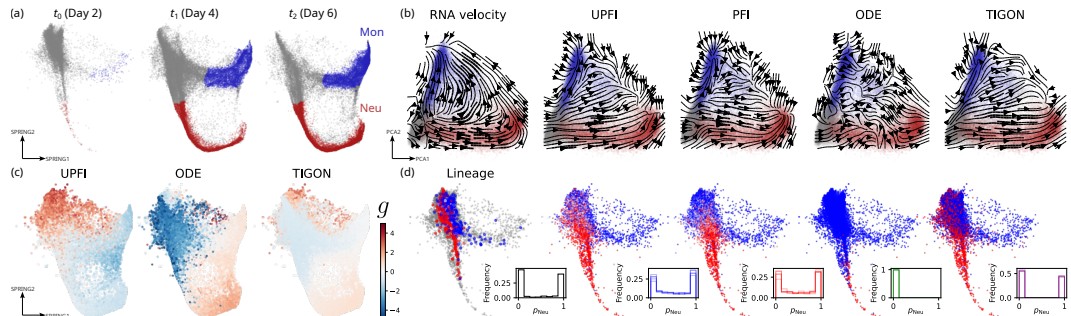

Figure 5: **Monocyte-neutrophil development.** (a) Temporal snapshots of monocyte-neutrophil fate determination, shown using SPRING coordinates and celltype annotations from the original publication. (b) Learned vector fields: RNA velocity vector field learned from spliced-unspliced data, all others from temporal snapshots. Cosine distance (relative to RNA velocity field) for Mon, Neu cells shown. (c) Learned growth rates. (d) Fate probabilities empirically estimated from lineage tracing data and predicted from learned dynamics. Pearson correlation (relative to lineage tracing data) shown.

tems are of the form

$$\mathrm{d}\boldsymbol{X}_t = (\boldsymbol{u}(\boldsymbol{X}_t) - \boldsymbol{v}(\boldsymbol{X}_t))\,\mathrm{d}t + \tfrac{1}{\sqrt{V}}\sqrt{\boldsymbol{u}(\boldsymbol{X}_t) + \boldsymbol{v}(\boldsymbol{X}_t)}\,\mathrm{d}\boldsymbol{B}_t. \tag{16}$$

where $\boldsymbol{u}, \boldsymbol{v}$ specify production and degradation rates and $V > 0$ is the system volume. In particular, the CLE involves a *multiplicative* noise model where the diffusivity varies with $\boldsymbol{x}$. This is in contrast to the additive noise setting of Fig. 3 where the diffusivity is fixed. For the bifurcating system, we specify a birth rate $b_t(\boldsymbol{x})$ causing cells in one branch to divide faster than others. In the HSC system, we specify $b_t$ similarly so that cells in the Erythrocyte branch divide faster. We then apply UPFI and PFI: for each inference method, we consider both *additive* and *multiplicative* noise models

$$\mathrm{d}\boldsymbol{X}_t = \boldsymbol{v}_\theta(\boldsymbol{x})\,\mathrm{d}t + \sigma\,\mathrm{d}\boldsymbol{B}_t, \tag{add.}$$

$$\mathrm{d}\boldsymbol{X}_t = (\boldsymbol{f}_\theta(\boldsymbol{x}) - \boldsymbol{g}_\theta(\boldsymbol{x}))\,\mathrm{d}t + \sigma\sqrt{\boldsymbol{f}_\theta(\boldsymbol{x}) + \boldsymbol{g}_\theta(\boldsymbol{x})}\,\mathrm{d}\boldsymbol{B}_t. \tag{mult.}$$

We show in Fig. 4(a, b)(i) the learned vector field for the bifurcating and HSC systems for each method and model, and highlight fate probabilities towards the fast growing branch. We find PFI in general infers a spurious force towards the fast-growing branch as in the previous example and consequently incorrect fate probabilities. On the other hand, UPFI with both additive and multiplicative noise models produce qualitatively similar results that resemble the ground truth. Quantitatively we confirm this in Table 2, and also find that UPFI with the multiplicative noise model mostly outperforms the additive noise model by a small but consistent margin.

Since UPFI and PFI work with a neural parameterisation of the force $\boldsymbol{v}$, this allows for plug-in use of interpretable architectures. To illustrate this, we make use of the Neural Graphical Model (NGM) architecture [42] from which a sparse and unsigned "causal graph" can be extracted and interpreted as the learned regulatory network. For simplicity we use the additive noise model, although use of NGM in the multiplicative noise case is also straightforward. We show in Fig. 4(a,b)(ii) the learned adjacency matrices along with the true interactions highlighted in red, and (iii) the precision-recall curves quantifying the network inference accuracy. In both cases UPFI recovers a more accurate network than PFI – see Table 3 for full evaluation results. This highlights that modelling of growth is essential to making accurate inferences on dynamics and thus gene interactions [2, 19], as well as the possibility for incorporating custom architectures within the UPFI approach.

**Lineage-tracing single-cell RNA-seq dataset** We now apply UPFI to an experimental time-course of monocyte-neutrophil development in vitro [1] across 3 timepoints over 4 days (Fig. 5). Additional information on the dynamics are available in two forms: (i) lineage tracing, where descendants of a common ancestor cell are distinguished by unique molecular labels ("barcodes"), and (ii) "RNA velocity" information [43], which provides partial information on the gene expression dynamics from relative abundances of spliced and unspliced RNA transcripts. This information, although limited, is independent from the population snapshots and so provides a point of comparison for the dynamics inferred from snapshots. We preprocess the data following standard pipelines

| $t$ | Celltype | UPFI | PFI | FITNESS-ODE | TIGON++ |
|---|---|---|---|---|---|
| | | | | **Cosine distance ($\downarrow$)** | |
| 0 | Monocyte | **0.26 $\pm$ 0.01** | 0.33 $\pm$ 0.02 | 0.42 $\pm$ 0.04 | 0.30 $\pm$ 0.02 |
| 0 | Neutrophil | 0.27 $\pm$ 0.02 | 0.31 $\pm$ 0.01 | 0.44 $\pm$ 0.02 | **0.26 $\pm$ 0.01** |
| 1 | Monocyte | **0.26 $\pm$ 0.01** | 0.32 $\pm$ 0.01 | 0.46 $\pm$ 0.04 | 0.28 $\pm$ 0.02 |
| 1 | Neutrophil | 0.24 $\pm$ 0.01 | 0.28 $\pm$ 0.01 | 0.43 $\pm$ 0.01 | **0.22 $\pm$ 0.01** |
| 2 | Monocyte | **0.27 $\pm$ 0.01** | 0.36 $\pm$ 0.01 | 0.41 $\pm$ 0.03 | 0.33 $\pm$ 0.02 |
| 2 | Neutrophil | 0.31 $\pm$ 0.01 | 0.33 $\pm$ 0.01 | 0.38 $\pm$ 0.01 | **0.27 $\pm$ 0.01** |
| | Metric | | | **Fate correlation ($\uparrow$)** | |
| | Kendall's $\tau$ | **0.22 $\pm$ 0.03** | 0.07 $\pm$ 0.03 | -0.02 $\pm$ 0.01 | 0.19 $\pm$ 0.04 |
| | Pearson's $r$ | **0.26 $\pm$ 0.04** | 0.09 $\pm$ 0.04 | -0.02 $\pm$ 0.01 | 0.19 $\pm$ 0.05 |
| | Spearman's $\rho$ | **0.27 $\pm$ 0.04** | 0.08 $\pm$ 0.04 | -0.03 $\pm$ 0.01 | 0.20 $\pm$ 0.05 |

Table 4: Evaluation results on lineage tracing dataset [1].

[43, 44] and use the leading 10 principal components to represent cell state. We apply UPFI with an additive noise model as well as several other methods to this data and show results for the learned drift alongside the RNA velocity field in Fig. 5(b). Since the magnitude of RNA velocity estimates are unreliable [45] we measure the cosine similarity between the RNA velocity field and the inferred force from snapshots (Table 4). We restrict this analysis to the monocyte and neutrophil cell clusters, corresponding to cells that are already committed to their respective lineages. We find that UPFI and TIGON++ perform comparably.

Inferred growth rates (Fig. 5(c)) show that UPFI predicts that cells in the earlier progenitor states have a higher division rate, which is consistent with the expected biology [1, 24]. On the other hand, the fitness-ODE yields the opposite while TIGON++ predicts a very low growth rate for most cells. The availability of lineage tracing data across timepoints provide us with a ground truth for measuring accuracy of fate predictions (Fig. 5): fate probabilities predicted by UPFI agree with the lineage tracing data more closely than other methods. In particular, PFI and fitness-ODE exhibit a bias towards monocyte lineage. While TIGON++ is able to predict a mixture of both fates, as in the bistable system it produces only deterministic outcomes. Similarly, quantitative results (Table 4) show that UPFI performs best in several different fate correlation metrics.

Finally, we note that this dataset was also studied in [24, 27] which introduced TIGON and Deep-RUOT respectively. However, in these papers all analyses of this dataset were carried out using two-dimensional embeddings computed by a force-directed layout algorithm [1]. The setting we consider (10-dimensional PCA embeddings) is more challenging, and also a more realistic scenario since nonlinear dimensionality reduction methods are known to introduce distortions and are thus unsuitable for downstream quantitative analysis [46].

# 4 Discussion

We study the problem of inferring stochastic dynamics from cross-sectional snapshots of a population subject to drift, diffusion as well as birth and death. To this end we propose UPFI, a method based on the probability flow characterisation of the Fokker-Planck equation. Our approach is flexible enough to handle generic noise models as well as growth, and crucially applies to the practically important scenario where growth rates are unknown and must be learned together with the drift. We provide a theoretical treatment in the linear-quadratic case, demonstrating the nature of the non-identifiability that arises from growth. We conclude by showcasing our method's efficacy using both simulated and experimental single cell data.

There are overall several lessons to be summarised from our work, which may be of relevance more generally to the problems of reconstructing dynamics from snapshots. As shown by our theoretical analysis and supported by results, in the fully general setting where the drift and growth components are non-autonomous, there is little reason to expect to accurately separate drift from growth effects. In the absence of additional information to aid with inference, we advocate to impose an autonomous drift and possibly also autonomous growth field. This is consistent with cell-autonomous models of biological dynamics, i.e. ignoring cell-cell interactions or temporally varying environments. Subject to the modelling limitations incurred by these assumptions, we argue that this serves as a strong inductive bias that can help inference.

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

# A  General results for Ornstein-Uhlenbeck processes

## A.1  OU process with affine fitness

**Proposition A.1.** *Consider an OU process with affine fitness, whose density satisfies*

$$\partial_t p_t = -\nabla \cdot ((\boldsymbol{A}_t \boldsymbol{x} + \boldsymbol{e}_t) p_t - \boldsymbol{D}_t \nabla p_t) + (b_t + \boldsymbol{c}_t^\top \boldsymbol{x}) p_t. \tag{17}$$

*where $\boldsymbol{A}_t \in \mathbb{R}^{d \times d}$, $\boldsymbol{e}_t, \boldsymbol{c}_t \in \mathbb{R}^d$, and $b_t \in \mathbb{R}$ are general, $\boldsymbol{D}_t \in \mathbb{R}^{d \times d}$ is symmetric positive definite. If $\rho_0 = m_0 \mathcal{N}(\boldsymbol{\mu}_0, \boldsymbol{\Sigma}_0)$, then $\rho_t = m_t \mathcal{N}(\boldsymbol{\mu}_t, \boldsymbol{\Sigma}_t)$ for all $t \geqslant 0$, where*

$$\dot{\boldsymbol{\Sigma}}_t = \boldsymbol{A}_t \boldsymbol{\Sigma}_t + \boldsymbol{\Sigma}_t \boldsymbol{A}_t^\top + 2\boldsymbol{D}_t, \tag{18}$$

$$\dot{\boldsymbol{\mu}}_t = \boldsymbol{A}_t \boldsymbol{\mu}_t + \boldsymbol{\Sigma}_t \boldsymbol{c}_t + \boldsymbol{e}_t, \tag{19}$$

$$\frac{\dot{m}_t}{m_t} = \boldsymbol{c}_t^\top \boldsymbol{\mu}_t + b_t. \tag{20}$$

*Proof.* We denote $\widehat{p}_t(\boldsymbol{k})$ the Fourier transform of $p_t(\boldsymbol{x})$. We have the following identities

$$\mathcal{F}\left(\nabla \cdot (\boldsymbol{A}_t \boldsymbol{x} + \boldsymbol{e}_t) p_t\right) = -\boldsymbol{k}^\top \boldsymbol{A}_t \nabla_{\boldsymbol{k}} \widehat{p}_t + i \boldsymbol{e}_t^\top \boldsymbol{k} \widehat{p}_t, \tag{21}$$

$$\mathcal{F}\left(\nabla \cdot (\boldsymbol{D}_t \nabla p_t)\right) = -\boldsymbol{k}^\top \boldsymbol{D}_t \boldsymbol{k} \widehat{p}_t. \tag{22}$$

Such that the Fourier transform of the PDE reads

$$\partial_t \widehat{p}_t = (\boldsymbol{k}^\top \boldsymbol{A}_t + i \boldsymbol{c}_t^\top) \nabla_{\boldsymbol{k}} \widehat{p}_t - \boldsymbol{k}^\top \boldsymbol{D}_t \boldsymbol{k} \widehat{p}_t + (b_t - i \boldsymbol{e}_t^\top \boldsymbol{k}) \widehat{p}_t. \tag{23}$$

We define the characteristics as the solution of

$$\frac{d\boldsymbol{k}_s}{ds} = -\boldsymbol{A}_s^\top \boldsymbol{k}_s - i \boldsymbol{c}_s, \tag{24}$$

such that

$$\frac{d\widehat{p}_s(\boldsymbol{k}_s)}{ds} = \left(\frac{d\boldsymbol{k}_s}{ds}\right)^\top \nabla_{\boldsymbol{k}} \widehat{p}_s(\boldsymbol{k}_s) + \partial_s \widehat{p}_s(\boldsymbol{k}_s) \tag{25}$$

$$= -\boldsymbol{k}_s^\top \boldsymbol{D}_s \boldsymbol{k}_s \widehat{p}_s + (b_s - i \boldsymbol{e}_s^\top \boldsymbol{k}_s) \widehat{p}_s. \tag{26}$$

Denote $\Phi(t, t_0)$ the state transition matrix solution to the homogeneous ODE (24) with initial condition $t_0$. It exists because the coefficient are locally integrable on $\mathbb{R}^+$ [47]. The state transition matrix has the following properties [48]:

$$\frac{d\Phi(s, u)}{ds} = -\boldsymbol{A}_s^\top \Phi(s, u), \quad \frac{d\Phi(s, u)}{du} = \Phi(s, u) \boldsymbol{A}_s^\top, \tag{27}$$

$$\Phi(t, u)\Phi(u, s) = \Phi(t, s), \quad \Phi(t, s)^{-1} = \Phi(s, t), \quad \Phi(s, s) = \boldsymbol{I}. \tag{28}$$

Denote $\Phi(s, 0) = \Phi_s$ for simplicity, we have, using the initial condition $\boldsymbol{k}_0$:

$$\boldsymbol{k}_s = \Phi_s \boldsymbol{k}_0 - i \int_0^s \Phi(s, u) \boldsymbol{c}_u du. \tag{29}$$

We introduce $\widetilde{\boldsymbol{v}}_s = -\int_0^s \Phi(s, u) \boldsymbol{c}_u du$, such that we have the Fourier transform along the characteristics reads, with an initial Gaussian distribution

$$\widehat{p}_s(\boldsymbol{k}_s) = \exp\left[ -i\boldsymbol{k}_0^\top \boldsymbol{x}_0 - \frac{1}{2}\boldsymbol{k}_0^\top \boldsymbol{\Sigma}_0 \boldsymbol{k}_0 - \boldsymbol{k}_0^\top \int_0^s \Phi_u^\top \boldsymbol{D}_u \Phi_u du \boldsymbol{k}_0 - 2i\boldsymbol{k}_0^\top \int_0^s \Phi_u^\top \boldsymbol{D}_u \widetilde{\boldsymbol{v}}_u du \right.$$

$$\left. + \int_0^s \widetilde{\boldsymbol{v}}_u^\top \boldsymbol{D}_u \widetilde{\boldsymbol{v}}_u du + \int_0^s b_u du - i \int_0^s \boldsymbol{e}_u^\top \Phi_u \boldsymbol{k}_0 + \int_0^s \boldsymbol{e}_u^\top \widetilde{\boldsymbol{v}}_u du \right]. \tag{30}$$

We introduce the following quantities

$$\boldsymbol{\Psi}_s = \int_0^s \Phi_u^\top \boldsymbol{D}_u \Phi_u du, \quad \boldsymbol{q}_s^\top = \int_0^s \boldsymbol{e}_u^\top \Phi_u du, \quad \boldsymbol{u}_s = \int_0^s \Phi_u^\top \boldsymbol{D}_u \widetilde{\boldsymbol{v}}_u du, \tag{31}$$

$$\gamma_s = \int_0^s \widetilde{\boldsymbol{v}}_u^\top \boldsymbol{D}_u \widetilde{\boldsymbol{v}}_u du, \quad w_s = \int_0^s \boldsymbol{e}_u^\top \widetilde{\boldsymbol{v}}_u du. \tag{32}$$

Inverting the equation for characteristics we have $\boldsymbol{k}_0 = \Phi_s^{-1}\boldsymbol{k}_s + i\Phi_s^{-1}\int_0^s \Phi(s,u)\boldsymbol{c}_u du = \Phi_s^{-1}\boldsymbol{k}_s - i\Phi_s^{-1}\widetilde{\boldsymbol{v}}_s$. We can directly see that the Fourier transform is quadratic in $\boldsymbol{k}$, such that it is the Fourier transform of a Gaussian measure. By considering one after the other the terms $O(k^2)$, $O(k)$ and $O(1)$, we can find analytical expressions for the covariance, the mean and the mass of the Gaussian measure as a function of the state transition matrix. They read:

$$\boldsymbol{\Sigma}_s = \Phi_s^{-T}\boldsymbol{\Sigma}_0\Phi_s^{-1} + 2\Phi_s^{-T}\boldsymbol{\Psi}_s\Phi_s^{-1}, \tag{33}$$

$$\boldsymbol{\mu}_s = \Phi_s^{-T}(\boldsymbol{\mu}_0 + 2\boldsymbol{u}_s + \boldsymbol{q}_s) - \boldsymbol{\Sigma}_s\widetilde{\boldsymbol{v}}_s, \tag{34}$$

$$\log m_s = -\widetilde{\boldsymbol{v}}_s^\top \boldsymbol{\mu}_s - \frac{1}{2}\widetilde{\boldsymbol{v}}_s^\top \boldsymbol{\Sigma}_s \widetilde{\boldsymbol{v}}_s + w_s + \gamma_s + \int_0^s b_u du. \tag{35}$$

Taking derivatives of these quantities and using the properties of the state transition matrix we recover the expected results. $\qquad \square$

### A.2 OU process with quadratic fitness

**Proposition 2.1** (OU process with quadratic fitness). *Consider an OU process with quadratic fitness, whose density satisfies*

$$\partial_t \rho_t(\boldsymbol{x}) = -\boldsymbol{\nabla}\cdot((\boldsymbol{A}_t\boldsymbol{x} + \boldsymbol{e}_t)\rho_t(\boldsymbol{x}) - \boldsymbol{D}_t\boldsymbol{\nabla}\rho_t(\boldsymbol{x})) + (b_t + \boldsymbol{c}_t^\top\boldsymbol{x} + \frac{1}{2}\boldsymbol{x}^\top\boldsymbol{\Gamma}_t\boldsymbol{x})\rho_t(\boldsymbol{x}), \tag{8}$$

*where $\boldsymbol{A}_t \in \mathbb{R}^{d\times d}$, $\boldsymbol{e}_t, \boldsymbol{c}_t \in \mathbb{R}^d$, and $b_t \in \mathbb{R}$ are generic, $\boldsymbol{D}_t \in \mathbb{R}^{d\times d}$ is symmetric positive definite, and $\boldsymbol{\Gamma}_t \in \mathbb{R}^{d\times d}$ is symmetric negative semi-definite. If $\rho_0 = m_0\mathcal{N}(\boldsymbol{\mu}_0,\boldsymbol{\Sigma}_0)$ with $m_0 > 0$, then $\rho_t = m_t\mathcal{N}(\boldsymbol{\mu}_t,\boldsymbol{\Sigma}_t)$ for all $t \geqslant 0$, where*

$$\dot{\boldsymbol{\Sigma}}_t = \boldsymbol{A}_t\boldsymbol{\Sigma}_t + \boldsymbol{\Sigma}_t\boldsymbol{A}_t^\top + 2\boldsymbol{D}_t + \boldsymbol{\Sigma}_t\boldsymbol{\Gamma}_t\boldsymbol{\Sigma}_t, \tag{9}$$

$$\dot{\boldsymbol{\mu}}_t = (\boldsymbol{A}_t + \boldsymbol{\Sigma}_t\boldsymbol{\Gamma}_t)\boldsymbol{\mu}_t + \boldsymbol{\Sigma}_t\boldsymbol{c}_t + \boldsymbol{e}_t, \tag{10}$$

$$\frac{\dot{m}_t}{m_t} = \frac{1}{2}\boldsymbol{\mu}_t^\top\boldsymbol{\Gamma}_t\boldsymbol{\mu}_t + \boldsymbol{c}_t^\top\boldsymbol{\mu}_t + b_t + \frac{1}{2}\operatorname{tr}(\boldsymbol{\Gamma}_t\boldsymbol{\Sigma}_t). \tag{11}$$

*Proof.* Let's use the ansatz $p_t(\boldsymbol{x}) = u_t(\boldsymbol{x})\exp\left[\boldsymbol{x}^\top\boldsymbol{G}_t\boldsymbol{x}/2\right]$ where $\boldsymbol{G}_t$ is symmetric. The equation for $u_t$ now reads

$$\begin{aligned}
\partial_t u_t &+ \frac{1}{2}u_t\boldsymbol{x}^\top\frac{d\boldsymbol{G}_t}{dt}\boldsymbol{x} \\
&= -\boldsymbol{\nabla}\cdot((\boldsymbol{A}_t\boldsymbol{x} + \boldsymbol{e}_t - 2\boldsymbol{D}_t\boldsymbol{G}_t\boldsymbol{x})u_t - \boldsymbol{D}_t\boldsymbol{\nabla}u_t) \\
&\quad + \left(b_t - \operatorname{tr}(\boldsymbol{G}_t\boldsymbol{D}_t) + \boldsymbol{c}_t^\top\boldsymbol{x} - \boldsymbol{e}_t^\top\boldsymbol{G}_t\boldsymbol{x} + \boldsymbol{x}^\top(\frac{1}{2}\boldsymbol{\Gamma}_t + \boldsymbol{G}_t\boldsymbol{D}_t\boldsymbol{G}_t - \boldsymbol{A}_t^\top\boldsymbol{G}_t)\boldsymbol{x}\right)u_t.
\end{aligned} \tag{36}$$

The quadratic term vanishes when $\boldsymbol{G}_t$ verifies for all $\boldsymbol{x}$ the equation $\boldsymbol{x}^\top(\boldsymbol{\Gamma}_t/2 + \boldsymbol{G}_t\boldsymbol{D}_t\boldsymbol{G}_t - \boldsymbol{A}_t^\top\boldsymbol{G}_t)\boldsymbol{x} = \frac{1}{2}\boldsymbol{x}^\top\dot{\boldsymbol{G}}_t\boldsymbol{x}$. This is verified for $\boldsymbol{G}_t$ satisfying the matrix Riccati equation

$$\boldsymbol{\Gamma}_t + 2\boldsymbol{G}_t\boldsymbol{D}_t\boldsymbol{G}_t - \boldsymbol{A}_t^\top\boldsymbol{G}_t - \boldsymbol{G}_t\boldsymbol{A}_t = \frac{d\boldsymbol{G}_t}{dt}. \tag{37}$$

Provided the coefficients are locally integrable on $\mathbb{R}^+$, that $\boldsymbol{\Gamma}_t \leqslant 0$ and $\boldsymbol{D}_t \geqslant 0$, if $\boldsymbol{G}_0 \leqslant 0$, then there exists a unique solution $\boldsymbol{G}_t$ on $\mathbb{R}^+$ for (37), and $\boldsymbol{G}_t \leqslant 0$ for all $t \geqslant 0$ [49]. We consider this unique solution with $\boldsymbol{G}_0 = 0$. Then, the equation for $u_t$ becomes affine in growth

$$\partial_t u_t = -\boldsymbol{\nabla}\cdot((\boldsymbol{A}_t\boldsymbol{x} + \boldsymbol{e}_t - 2\boldsymbol{D}_t\boldsymbol{G}_t\boldsymbol{x})u_t - \boldsymbol{D}_t\boldsymbol{\nabla}u_t) + \left(b_t - \operatorname{tr}(\boldsymbol{G}_t\boldsymbol{D}_t) + \boldsymbol{c}_t^\top\boldsymbol{x} - \boldsymbol{e}_t^\top\boldsymbol{G}_t\boldsymbol{x}\right)u_t. \tag{38}$$

We redefine $\widetilde{\boldsymbol{A}}_t = \boldsymbol{A}_t - 2\boldsymbol{D}_t\boldsymbol{G}_t$, $\widetilde{b}_t = b_t - \operatorname{tr}(\boldsymbol{G}_t\boldsymbol{D}_t)$ and $\widetilde{\boldsymbol{c}}_t = \boldsymbol{c}_t - \boldsymbol{G}_t\boldsymbol{e}_t$. The equation now reads

$$\partial_t u_t = -\boldsymbol{\nabla}\cdot\left((\widetilde{\boldsymbol{A}}_t\boldsymbol{x} + \boldsymbol{e}_t)u_t - \boldsymbol{D}_t\boldsymbol{\nabla}u_t\right) + \left(\widetilde{b}_t + \widetilde{\boldsymbol{c}}_t^\top\boldsymbol{x}\right)u_t. \tag{39}$$

We can apply the results of Prop. A.1. Since $u_0(\boldsymbol{x}) = p_0(\boldsymbol{x})$ is a Gaussian measure, it remains Gaussian, and we denote its covariance $\widetilde{\boldsymbol{\Sigma}}_t$, its mean $\widetilde{\boldsymbol{x}}_t$ and its mass $\widetilde{m}_t$. It follows that $p_t(\boldsymbol{x})$ is

Gaussian measure for all $t \geqslant 0$ since $\widetilde{\Sigma}_t^{-1} - G_t$ is positive definite as the sum of positive definite and positive semi-definite terms. The covariance of this Gaussian measure is then $\Sigma_t = (\widetilde{\Sigma}_t^{-1} - G_t)^{-1}$.

We have the following fact for any differentiable matrix valued function $B_t$ invertible for all $t$:

$$\frac{dB_t^{-1}}{dt} = -B_t^{-1}\frac{dB_t}{dt}B_t^{-1}. \tag{40}$$

Applying this to the covariance, we find:

$$\frac{d\Sigma_t}{dt} = -\Sigma_t \frac{d\Sigma_t^{-1}}{dt}\Sigma_t = \Sigma_t \widetilde{\Sigma}_t^{-1}\left(\widetilde{A}_t\widetilde{\Sigma}_t + \widetilde{\Sigma}_t\widetilde{A}_t^\top + 2D_t\right)\widetilde{\Sigma}_t^{-1}\Sigma_t + \Sigma_t\frac{dG_t}{dt}\Sigma_t. \tag{41}$$

Substituting $\widetilde{A}_t = A_t - 2D_tG_t$ and $\dot{G}_t = \Gamma_t + 2G_tD_tG_t - A_t^\top G_t - G_tA_t$ we have

$$\frac{d\Sigma_t}{dt} = A_t\Sigma_t + \Sigma_tA_t^\top + 2D_t + \Sigma_t\Gamma_t\Sigma_t. \tag{42}$$

Completing the square in high-dimensions allows us to write the mean of the overall process as

$$\mu_t = \left(\widetilde{\Sigma}_t^{-1} - G_t\right)^{-1}\widetilde{\Sigma}_t^{-1}\widetilde{\mu}_t = \Sigma_t\widetilde{\Sigma}_t^{-1}\widetilde{\mu}_t = (I + \Sigma_tG_t)\widetilde{\mu}_t. \tag{43}$$

Using Prop. A.1 we have

$$\frac{dx_t}{dt} = (\frac{\Sigma_t}{dt}G_t + \Sigma_t\frac{dG_t}{dt})\widetilde{\mu}_t + (I + \Sigma_tG_t)(\widetilde{A}_t\widetilde{\mu}_t + \widetilde{\Sigma}_t\widetilde{c}_t + \widetilde{e}_t) \tag{44}$$

We replace $\widetilde{A}_t = A_t - 2D_tG_t$ and we use the Riccati equation in the terms multiplying $\widetilde{\mu}_t$. In the other terms we replace $\widetilde{c}_t = c_t - G_te_t$ and $e_t = \widetilde{e}_t$. We find:

$$\frac{d\mu_t}{dt} = (A_t + \Sigma_t\Gamma_t)\mu_t + \Sigma_tc_t + e_t. \tag{45}$$

We can derive the fitness following the same approach and using lengthy simplifications. We only need to gather together the terms remaining after completing the square, as well as adjust for the change in covariance in the normalisation factor. However, this result is more easily obtained by using a result for the mean over a Gaussian probability distribution of quadratic form [50, Theorem 1.5]. We find that

$$\frac{\dot{m}}{m} = \mathbb{E}[b_t + c_t^\top X + \frac{1}{2}X^\top\Gamma_tX] = b_t + c_t^\top\mu_t + \frac{1}{2}\text{tr}(\Gamma_t\Sigma_t) + \frac{1}{2}\mu_t^\top\Gamma_t\mu_t \tag{46}$$

where the expectation is understood to be taken with respect to $\mathcal{N}(x_t, \Sigma_t)$. $\qquad\square$

**Corollary 2.2.** *Consider an OU process with drift $v_t(x) = A_tx$ and a time-dependent fitness as in (8). Let $K_t \in \mathbb{R}^{n \times n}$ be an arbitrary matrix. Then there exists $\epsilon > 0$ such that the system is indistinguishable from another OU process with drift $v_t(x) = A_t + I + \epsilon K_t$ and time-dependent quadratic fitness.*

*Proof.* Let's define the following growth parameters: $\widetilde{b}_t = b_t - \text{tr}((I + \epsilon K_t)\Sigma_t)/2$, $\widetilde{c}_t = -(I + \epsilon K_t)^\top\Sigma_t^{-1}\mu_t$ and $\widetilde{\Gamma}_t = \Gamma_t - \big((I + \epsilon K_t)^\top\Sigma_t^{-1} + \Sigma_t^{-1}(I + \epsilon K_t)\big)$. With these and the drift $v_t(x)$, the solution to the system of ODE above is unchanged. We take $\epsilon$ as the largest value such that $\widetilde{\Gamma}_t$ is negative definite. This value is strictly larger than zero thanks to the identity. This derivation still holds if the drift is autonomous: if $v_t(x) = Ax$, then there for any $K$, there exists $\epsilon > 0$ such that the system is indistinguishable from another OU process with $\widetilde{v}_t(x) = A + I + \epsilon K$ and time-dependent quadratic fitness.

$\qquad\square$

# B  Continuous-time loss for OU processes with quadratic fitness

**Theorem 2.3** (Loss function for OU processes with quadratic fitness). *Consider the true process as well as the inferred processed to both be OU processes with quadratic fitness, i.e. (8). Denoting*

$q = 2/\gamma$, with $K\Delta t$ fixed, when $\Delta t \to 0$ we have $\Delta t^{-1} L \to \mathcal{L}$, where $\mathcal{L}$ is the continuous time loss function:

$$\mathcal{L} = q^{-1} \int_0^T \mathrm{d}t\, m_t \bigg( \|\boldsymbol{v}_t - \widehat{\boldsymbol{v}}_t\|_{\boldsymbol{X}_t^{-1}}^2 + \frac{1}{2}(h_t - \widehat{h}_t)^2 \tag{13}$$

$$+ q \sum_{i,j} \frac{\sigma_{i,t}^2 \sigma_{j,q,t}^2}{(\sigma_{j,q,t}^2 \sigma_{i,t}^2 + \sigma_{i,q,t}^2 \sigma_{j,t}^2)^2} \left( \boldsymbol{w}_{i,t}^\top (\boldsymbol{B}_t - \widehat{\boldsymbol{B}}_t) \boldsymbol{w}_{j,t} \right)^2 \bigg) + \lambda \int_0^T R_t\, \mathrm{d}t \tag{14}$$

where $\boldsymbol{\Sigma}_t = \sum_i \sigma_{i,t}^2 \boldsymbol{w}_{i,t} \boldsymbol{w}_{i,t}^\top$ is the eigendecomposition of the covariance $\boldsymbol{\Sigma}_t$, $\sigma_{i,q,t}^2 = 1 + q\sigma_{i,t}^2$ for all $i$, $\boldsymbol{X}_t = 2\boldsymbol{\Sigma}_t + q^{-1}$. We also have

$$\boldsymbol{B}_t - \widehat{\boldsymbol{B}}_t = \boldsymbol{\Sigma}_t (\boldsymbol{A}_t - \widehat{\boldsymbol{A}}_t)^\top + (\boldsymbol{A}_t - \widehat{\boldsymbol{A}}_t)\boldsymbol{\Sigma}_t + \boldsymbol{\Sigma}_t (\boldsymbol{\Gamma}_t - \widehat{\boldsymbol{\Gamma}}_t)\boldsymbol{\Sigma}_t + 2(\boldsymbol{D}_t - \widehat{\boldsymbol{D}}_t),$$

$$\boldsymbol{v}_t - \widehat{\boldsymbol{v}}_t = ((\boldsymbol{A}_t - \widehat{\boldsymbol{A}}_t) + \boldsymbol{\Sigma}_t(\boldsymbol{\Gamma}_t - \widehat{\boldsymbol{\Gamma}}_t))\boldsymbol{\mu}_t + \boldsymbol{\Sigma}_t(\boldsymbol{c}_t - \widehat{\boldsymbol{c}}_t) + (\boldsymbol{e}_t - \widehat{\boldsymbol{e}}_t), \tag{15}$$

$$h_t - \widehat{h}_t = \frac{1}{2}\boldsymbol{\mu}_t^\top (\boldsymbol{\Gamma}_t - \widehat{\boldsymbol{\Gamma}}_t)\boldsymbol{\mu}_t + (\boldsymbol{c}_t - \widehat{\boldsymbol{c}}_t)^\top \boldsymbol{\mu}_t + (b_t - \widehat{b}_t) + \frac{1}{2}\mathrm{tr}((\boldsymbol{\Gamma}_t - \widehat{\boldsymbol{\Gamma}}_t)\boldsymbol{\Sigma}_t),$$

and $R_t$ is a strongly convex function of the parameters defining the fitness and the drift. For $\lambda > 0$, $\mathcal{L}$ has a unique minimiser $t \mapsto (\widehat{\boldsymbol{A}}_t^\star, \widehat{\boldsymbol{e}}_t^\star, \widehat{b}_t^\star, \widehat{\boldsymbol{c}}_t^\star, \widehat{\boldsymbol{\Gamma}}_t^\star)$.

*Proof.* When the entropic regularisation $\epsilon = 0$, the unbalanced Sinkhorn divergence between two Gaussian measures reduces to the Gaussian-Hellinger-Kantorovich distance $S_{0,\gamma} = \mathrm{GHK}_\gamma$. Between the inferred and the true process at time $t_i$ it reads:

$$\mathrm{GHK}\left( m_{t_i}\mathcal{N}(\boldsymbol{\mu}_{t_i}, \boldsymbol{\Sigma}_{t_i}), \widehat{m}_{t_i}\mathcal{N}(\widehat{\boldsymbol{\mu}}_{t_i}, \widehat{\boldsymbol{\Sigma}}_{t_i}) \right)$$

$$= 2q^{-1}\left( m_{t_i} + \widehat{m}_{t_i} - 2\sqrt{\frac{m_{t_i}\widehat{m}_{t_i}}{\det \boldsymbol{J}}} \exp\left[ -\frac{1}{2}(\boldsymbol{\mu}_{t_i} - \widehat{\boldsymbol{\mu}}_{t_i})^\top \boldsymbol{X}_{t_i}^{-1}(\boldsymbol{\mu}_{t_i} - \widehat{\boldsymbol{\mu}}_{t_i}) \right] \right), \tag{47}$$

where we have

$$\boldsymbol{X}_{t_i} = \boldsymbol{\Sigma}_{t_i} + \boldsymbol{\Sigma}_{t_i} + q^{-1}\boldsymbol{I}, \ \boldsymbol{J} = (\boldsymbol{\Sigma}_{t_i,q}\boldsymbol{\Sigma}_{t_i,q})^{1/2}(\boldsymbol{I} - q(\boldsymbol{\Sigma}_{t_i}\boldsymbol{\Sigma}_{t_i,q}^{-1}\boldsymbol{\Sigma}_{t_i,q}^{-1}\boldsymbol{\Sigma}_{t_i})^{1/2}), \tag{48}$$

with

$$\boldsymbol{\Sigma}_{t_i,q} = q\boldsymbol{\Sigma}_{t_i} + \boldsymbol{I}, \ \boldsymbol{\Sigma}_{t_i,q} = q\boldsymbol{\Sigma}_{t_i} + \boldsymbol{I}. \tag{49}$$

Without loss of generality we consider the case $t_1 = \Delta t$, and we perform Taylor expansion in $\Delta t$ of the GHK loss. Because the final expansion will be of order $\Delta t^2$, only the terms of order $\Delta t$ of the covariances play a role.

The hard part in this expansion is the expansion of $\det\boldsymbol{J}$. We denote $\boldsymbol{\Sigma}_{\Delta t} = \boldsymbol{\Sigma}_0 + \Delta t\boldsymbol{A}$ and $\widetilde{\boldsymbol{\Sigma}}_{\Delta t} = \boldsymbol{\Sigma}_0 + \Delta t\widetilde{\boldsymbol{A}}$ and $\boldsymbol{\Sigma}_{0,q} = \boldsymbol{I} + q\boldsymbol{\Sigma}_0$. For this, we need to compute $\boldsymbol{\Sigma}_{\Delta t}\boldsymbol{\Sigma}_{\Delta t,q}^{-1}\widetilde{\boldsymbol{\Sigma}}_{\Delta t}\widetilde{\boldsymbol{\Sigma}}_{\Delta t,q}^{-1}$, up to second order in $\Delta t$. We have that

$$\boldsymbol{\Sigma}_{\Delta t}\boldsymbol{\Sigma}_{\Delta t,q}^{-1} = (\boldsymbol{\Sigma}_0 + \Delta t\boldsymbol{A})(\boldsymbol{\Sigma}_{0,q}^{-1} - q\Delta t\boldsymbol{\Sigma}_{0,q}^{-1}\boldsymbol{A}\boldsymbol{\Sigma}_{0,q}^{-1} + q^2\Delta t^2\boldsymbol{\Sigma}_{0,q}^{-1}\boldsymbol{A}\boldsymbol{\Sigma}_{0,q}^{-1}\boldsymbol{A}\boldsymbol{\Sigma}_{0,q}^{-1}) \tag{50}$$

$$= \boldsymbol{\Sigma}_0\boldsymbol{\Sigma}_{0,q}^{-1} + \Delta t(\boldsymbol{A}\boldsymbol{\Sigma}_{0,q}^{-1} - q\boldsymbol{\Sigma}_0\boldsymbol{\Sigma}_{0,q}^{-1}\boldsymbol{A}\boldsymbol{\Sigma}_{0,q}^{-1}) \tag{51}$$

$$+ \Delta t^2 q(q\boldsymbol{\Sigma}_0\boldsymbol{\Sigma}_{0,q}^{-1}\boldsymbol{A}\boldsymbol{\Sigma}_{0,q}^{-1}\boldsymbol{A}\boldsymbol{\Sigma}_{0,q}^{-1} - \boldsymbol{A}\boldsymbol{\Sigma}_{0,q}^{-1}\boldsymbol{A}\boldsymbol{\Sigma}_{0,q}^{-1}). \tag{52}$$

We also have

$$\widetilde{\boldsymbol{\Sigma}}_{\Delta t,q}^{-1}\widetilde{\boldsymbol{\Sigma}}_{\Delta t} = (\boldsymbol{\Sigma}_{0,q}^{-1} - q\Delta t\boldsymbol{\Sigma}_{0,q}^{-1}\widetilde{\boldsymbol{A}}\boldsymbol{\Sigma}_{0,q}^{-1} + q^2\Delta t^2\boldsymbol{\Sigma}_{0,q}^{-1}\widetilde{\boldsymbol{A}}\boldsymbol{\Sigma}_{0,q}^{-1}\widetilde{\boldsymbol{A}}\boldsymbol{\Sigma}_{0,q}^{-1})(\boldsymbol{\Sigma}_0 + \Delta t\widetilde{\boldsymbol{A}}) \tag{53}$$

$$= \boldsymbol{\Sigma}_{0,q}^{-1}\boldsymbol{\Sigma}_0 + \Delta t(\boldsymbol{\Sigma}_{0,q}^{-1}\widetilde{\boldsymbol{A}} - q\boldsymbol{\Sigma}_{0,q}^{-1}\widetilde{\boldsymbol{A}}\boldsymbol{\Sigma}_{0,q}^{-1}\boldsymbol{\Sigma}_0) \tag{54}$$

$$+ \Delta t^2 q(q\boldsymbol{\Sigma}_{0,q}^{-1}\widetilde{\boldsymbol{A}}\boldsymbol{\Sigma}_{0,q}^{-1}\widetilde{\boldsymbol{A}}\boldsymbol{\Sigma}_{0,q}^{-1}\boldsymbol{\Sigma}_0 - \boldsymbol{\Sigma}_{0,q}^{-1}\widetilde{\boldsymbol{A}}\boldsymbol{\Sigma}_{0,q}^{-1}\widetilde{\boldsymbol{A}}). \tag{55}$$

We can now gather the $O(1)$, $O(\Delta t)$, $O(\Delta t^2)$ separately. We denote them respectively $\boldsymbol{M}, \boldsymbol{H}_1, \boldsymbol{H}_{2345}$. We define the operation 'tt.' as the 'tilde transpose' operation, which is applied

to the term directly to its left. We therefore have

$$\boldsymbol{M} = \boldsymbol{\Sigma}_0 \boldsymbol{\Sigma}_{0,q}^{-2} \boldsymbol{\Sigma}_0 \tag{56}$$

$$\boldsymbol{H}_1 = \boldsymbol{\Sigma}_0 (\boldsymbol{\Sigma}_{0,q}^{-2} \widetilde{\boldsymbol{A}} - q\boldsymbol{\Sigma}_{0,q}^{-2} \widetilde{\boldsymbol{A}} \boldsymbol{\Sigma}_{0,q}^{-1} \boldsymbol{\Sigma}_0) + \text{tt.} \tag{57}$$

$$\boldsymbol{H}_{2345} = q\boldsymbol{\Sigma}_0 (q\boldsymbol{\Sigma}_{0,q}^{-2} \widetilde{\boldsymbol{A}} \boldsymbol{\Sigma}_{0,q}^{-1} \widetilde{\boldsymbol{A}} \boldsymbol{\Sigma}_{0,q}^{-1} \boldsymbol{\Sigma}_0 - \boldsymbol{\Sigma}_{0,q}^{-2} \widetilde{\boldsymbol{A}} \boldsymbol{\Sigma}_{0,q}^{-1} \widetilde{\boldsymbol{A}}) + \text{tt.} \tag{58}$$

$$+ \boldsymbol{A} \boldsymbol{\Sigma}_{0,q}^{-2} \widetilde{\boldsymbol{A}} + q^2 \boldsymbol{\Sigma}_0 \boldsymbol{\Sigma}_{0,q}^{-1} \boldsymbol{A} \boldsymbol{\Sigma}_{0,q}^{-2} \widetilde{\boldsymbol{A}} \boldsymbol{\Sigma}_{0,q}^{-1} \boldsymbol{\Sigma}_0 \tag{59}$$

$$- q\boldsymbol{A} \boldsymbol{\Sigma}_{0,q}^{-2} \widetilde{\boldsymbol{A}} \boldsymbol{\Sigma}_{0,q}^{-1} \boldsymbol{\Sigma}_0 + \text{tt.} \tag{60}$$

Using the Woodbury formula we also have $\boldsymbol{\Sigma}_0 \boldsymbol{\Sigma}_{0,q}^{-1} = q^{-1}(\boldsymbol{I} - \boldsymbol{\Sigma}_{0,q}^{-1})$. We have that $\boldsymbol{\Sigma}_0$ and $\boldsymbol{\Sigma}_{0,q}^{-1}$ commute, and that $\boldsymbol{M}^{1/2} = \boldsymbol{\Sigma}_0 \boldsymbol{\Sigma}_{0,q}^{-1} = q^{-1}(\boldsymbol{I} - \boldsymbol{\Sigma}_{0,q}^{-1})$. We introduce $\boldsymbol{S}(\boldsymbol{Y})$ the unique solution $\boldsymbol{X}$ to the following Lyapunov equation

$$\boldsymbol{M}^{1/2} \boldsymbol{X} + \boldsymbol{X} \boldsymbol{M}^{1/2} = \boldsymbol{Y} \tag{61}$$

This solution is expressed in terms of an integral (and is linear in $\boldsymbol{Y}$), ie.

$$\boldsymbol{S}(\boldsymbol{Y}) = \int_0^\infty e^{-\boldsymbol{M}^{1/2}t} \boldsymbol{Y} e^{-\boldsymbol{M}^{1/2}t} dt \tag{62}$$

As a result, we have

$$\left( \boldsymbol{\Sigma}_{\Delta t} \boldsymbol{\Sigma}_{\Delta t,q}^{-1} \widetilde{\boldsymbol{\Sigma}}_{\Delta t,q}^{-1} \widetilde{\boldsymbol{\Sigma}}_{\Delta t} \right)^{1/2} = \boldsymbol{M}^{1/2} + \Delta t \boldsymbol{S}(\boldsymbol{H}_1) + \Delta t^2 \left( \boldsymbol{S}(\boldsymbol{H}_{2345}) - \boldsymbol{S}(\boldsymbol{S}(\boldsymbol{H}_1)^2) \right) \tag{63}$$

We have $\boldsymbol{I} - q\boldsymbol{M}^{1/2} = \boldsymbol{\Sigma}_{0,q}^{-1}$, such that

$$\det \left( \boldsymbol{I} - q \left( \boldsymbol{\Sigma}_{\Delta t} \boldsymbol{\Sigma}_{\Delta t,q}^{-1} \widetilde{\boldsymbol{\Sigma}}_{\Delta t,q}^{-1} \widetilde{\boldsymbol{\Sigma}}_{\Delta t} \right)^{1/2} \right) = \det \boldsymbol{\Sigma}_{0,q}^{-1} \det \left( \boldsymbol{I} + \Delta t \boldsymbol{L} + \Delta t^2 \boldsymbol{G} \right), \tag{64}$$

where $\boldsymbol{L} = -q\boldsymbol{\Sigma}_{0,q} \boldsymbol{S}(\boldsymbol{H}_1)$ and $\boldsymbol{G} = -q\boldsymbol{\Sigma}_{0,q} \left( \boldsymbol{S}(\boldsymbol{H}_{2346}) - \boldsymbol{S}(\boldsymbol{S}(\boldsymbol{H}_1)^2) \right)$. We then have the following expansion at second order in $\Delta t$

$$\det \left( \boldsymbol{I} + \Delta t \boldsymbol{L} + \Delta t^2 \boldsymbol{G} \right) = 1 + \Delta t \operatorname{tr} \boldsymbol{L} + \frac{\Delta t^2}{2} \left( \operatorname{tr}^2 \boldsymbol{L} - \operatorname{tr} \boldsymbol{L}^2 + 2\operatorname{tr} \boldsymbol{G} \right). \tag{65}$$

So we need to compute $\operatorname{tr} \boldsymbol{L}$, $\operatorname{tr} \boldsymbol{G}$, $\operatorname{tr} \boldsymbol{L}^2$. We have, using the fact that $\boldsymbol{\Sigma}_{0,q}$ commutes with $\boldsymbol{M}^{1/2}$,

$$\operatorname{tr} \boldsymbol{L} = -q\operatorname{tr}(\boldsymbol{\Sigma}_{0,q} \int_0^\infty e^{-\boldsymbol{M}^{1/2}t} \boldsymbol{H}_1 e^{-\boldsymbol{M}^{1/2}t} dt) = -\frac{q}{2} \operatorname{tr}(\boldsymbol{M}^{-1/2} \boldsymbol{\Sigma}_{0,q} \boldsymbol{H}_1) \tag{66}$$

$$= -\frac{q}{2} \operatorname{tr}(\boldsymbol{\Sigma}_{0,q}^2 \boldsymbol{\Sigma}_0^{-1} \boldsymbol{H}_1) = -\frac{q}{2} \left( \operatorname{tr}(\boldsymbol{A} + \widetilde{\boldsymbol{A}}) - q\operatorname{tr}((\boldsymbol{A} + \widetilde{\boldsymbol{A}}) \boldsymbol{\Sigma}_{0,q}^{-1} \boldsymbol{\Sigma}_0) \right) \tag{67}$$

$$= -\frac{q}{2} \operatorname{tr}(\boldsymbol{\Sigma}_{0,q}^{-1}(\boldsymbol{A} + \widetilde{\boldsymbol{A}})). \tag{68}$$

**Computation of $\operatorname{tr} \boldsymbol{G}$**

Using the same trick as for $\operatorname{tr} \boldsymbol{L}$ and standard properties of the trace we have

$$\operatorname{tr}(-q\boldsymbol{\Sigma}_{0,q} \boldsymbol{S}(\boldsymbol{H}_{2345})) = \frac{1}{2} \left( q^2 \operatorname{tr}(\boldsymbol{A} \boldsymbol{\Sigma}_{0,q}^{-1} \boldsymbol{A} \boldsymbol{\Sigma}_{0,q}^{-1} + \text{tt.}) - q\operatorname{tr}(\boldsymbol{\Sigma}_0^{-1} \boldsymbol{A} \boldsymbol{\Sigma}_{0,q}^{-2} \widetilde{\boldsymbol{A}}) \right) \tag{69}$$

To compute $\operatorname{tr}(\boldsymbol{\Sigma}_{0,q} \boldsymbol{S}(\boldsymbol{S}(\boldsymbol{H}_1)^2))$ we denote $\boldsymbol{M}^{1/2} = \sum_i u_i \boldsymbol{w}_i \boldsymbol{w}_i^\top$ the eigendecomposition of $\boldsymbol{M}^{1/2}$. We also denote $\boldsymbol{W}_i = \boldsymbol{w}_i \boldsymbol{w}_i^\top$. Therefore we have

$$\boldsymbol{S}(\boldsymbol{H}_1) = \sum_{i,j} \frac{1}{u_i + u_j} \boldsymbol{W}_i \boldsymbol{H}_1 \boldsymbol{W}_i \tag{70}$$

such that

$$\boldsymbol{\Sigma}_{0,q} \boldsymbol{S}(\boldsymbol{S}(\boldsymbol{H}_1)^2) = \sum_{i,j,k} \frac{(1 - u_k q)^2 (1 - u_j q)}{(u_i + u_k)(u_j + u_k)(u_i + u_j)} \boldsymbol{W}_i \left( u_i \widetilde{\boldsymbol{A}} + u_k \boldsymbol{A} \right) \boldsymbol{W}_k \left( u_k \widetilde{\boldsymbol{A}} + u_j \boldsymbol{A} \right) \boldsymbol{W}_j. \tag{71}$$

Taking the trace we have

$$\text{tr}(\boldsymbol{\Sigma}_{0,q}\boldsymbol{S}(\boldsymbol{S}(\boldsymbol{H}_1)^2)) = \sum_{i,j} \frac{(1-u_jq)^2(1-u_iq)}{2u_i(u_i+u_j)^2} \text{tr}\left(\boldsymbol{W}_i(u_i\widetilde{\boldsymbol{A}}+u_j\boldsymbol{A})\boldsymbol{W}_j(u_j\widetilde{\boldsymbol{A}}+u_i\boldsymbol{A})\right). \quad (72)$$

Additionally, we have that

$$\text{tr}\left(\boldsymbol{W}_i(u_i\widetilde{\boldsymbol{A}}+u_j\boldsymbol{A})\boldsymbol{W}_j(u_j\widetilde{\boldsymbol{A}}+u_i\boldsymbol{A})\right) \quad (73)$$

$$= u_iu_j\text{tr}\left(\boldsymbol{W}_i(\boldsymbol{A}-\widetilde{\boldsymbol{A}})\boldsymbol{W}_j(\boldsymbol{A}-\widetilde{\boldsymbol{A}})\right) + (u_i+u_j)^2\text{tr}\left(\boldsymbol{W}_i\boldsymbol{A}\boldsymbol{W}_j\widetilde{\boldsymbol{A}}\right) \quad (74)$$

As a result we have

$$\text{tr}\boldsymbol{G} = \frac{1}{2}q^2\text{tr}(\boldsymbol{A}\boldsymbol{\Sigma}_{0,q}^{-1}\boldsymbol{A}\boldsymbol{\Sigma}_{0,q}^{-1} + \text{tt.}) + \frac{q}{2}\sum_{i,j}\frac{(1-u_jq)^2(1-u_iq)u_j}{(u_i+u_j)^2}\text{tr}\left(\boldsymbol{W}_i(\boldsymbol{A}-\widetilde{\boldsymbol{A}})\boldsymbol{W}_j(\boldsymbol{A}-\widetilde{\boldsymbol{A}})\right) \quad (75)$$

**Computation of $\text{tr}\boldsymbol{L}^2$**

Similarly we have

$$\boldsymbol{\Sigma}_{0,q}\boldsymbol{S}(\boldsymbol{H}_1) = \sum_{i,j}\frac{(1-u_jq)}{(u_i+u_j)}\boldsymbol{W}_i\left(u_i\widetilde{\boldsymbol{A}}+u_j\boldsymbol{A}\right)\boldsymbol{W}_j. \quad (76)$$

Using the same approach for $\text{tr}\boldsymbol{G}$ we find

$$\text{tr}((\boldsymbol{\Sigma}_{0,q}\boldsymbol{S}(\boldsymbol{H}_1))^2) = \sum_{i,j}\frac{(1-u_iq)(1-u_jq)u_iu_j}{(u_i+u_j)^2}\text{tr}\left(\boldsymbol{W}_i(\boldsymbol{A}-\widetilde{\boldsymbol{A}})\boldsymbol{W}_j(\boldsymbol{A}-\widetilde{\boldsymbol{A}})\right) \quad (77)$$

$$+ \text{tr}(\boldsymbol{\Sigma}_{0,q}^{-1}\boldsymbol{A}\boldsymbol{\Sigma}_{0,q}^{-1}\widetilde{\boldsymbol{A}}). \quad (78)$$

As a result we have

$$\text{tr}\boldsymbol{L}^2 = q^2\text{tr}(\boldsymbol{\Sigma}_{0,q}^{-1}\boldsymbol{A}\boldsymbol{\Sigma}_{0,q}^{-1}\widetilde{\boldsymbol{A}}) + q^2\sum_{i,j}\frac{(1-u_iq)(1-u_jq)u_iu_j}{(u_i+u_j)^2}\text{tr}\left(\boldsymbol{W}_i(\boldsymbol{A}-\widetilde{\boldsymbol{A}})\boldsymbol{W}_j(\boldsymbol{A}-\widetilde{\boldsymbol{A}})\right) \quad (79)$$

**Computation of $\det(\boldsymbol{\Sigma}_{\Delta t,q}\widetilde{\boldsymbol{\Sigma}}_{\Delta t,q})^{1/2}$**

We have

$$\boldsymbol{\Sigma}_{\Delta t,q}\widetilde{\boldsymbol{\Sigma}}_{\Delta t,q} = \boldsymbol{\Sigma}_{0,q}^2\left(\boldsymbol{I} + q\Delta t(\boldsymbol{\Sigma}_{0,q}^{-2}\boldsymbol{A}\boldsymbol{\Sigma}_{0,q} + \boldsymbol{\Sigma}_{0,q}^{-1}\widetilde{\boldsymbol{A}}) + q^2\Delta t^2\boldsymbol{\Sigma}_{0,q}^{-2}\boldsymbol{A}\widetilde{\boldsymbol{A}}\right). \quad (80)$$

Taking the determinant and Taylor expanding we have

$$\det(\boldsymbol{\Sigma}_{\Delta t,q}\widetilde{\boldsymbol{\Sigma}}_{\Delta t,q}) = \det(\boldsymbol{\Sigma}_{0,q})^2 \quad (81)$$

$$\times \left(1 + q\Delta t\text{tr}(\boldsymbol{\Sigma}_{0,q}^{-1}(\boldsymbol{A}+\widetilde{\boldsymbol{A}})) + q^2\frac{\Delta t^2}{2}\left((\text{tr}^2(\boldsymbol{\Sigma}_{0,q}^{-1}(\boldsymbol{A}+\widetilde{\boldsymbol{A}})) - \text{tr}(\boldsymbol{\Sigma}_{0,q}^{-1}\boldsymbol{A}\boldsymbol{\Sigma}_{0,q}^{-1}\boldsymbol{A} + \text{tt.})\right)\right). \quad (82)$$

Expanding the square root we find

$$\det(\boldsymbol{\Sigma}_{\Delta t,q}\widetilde{\boldsymbol{\Sigma}}_{\Delta t,q})^{1/2} = \det(\boldsymbol{\Sigma}_{0,q}) \quad (83)$$

$$\times \left(1 + q\frac{\Delta t}{2}\text{tr}(\boldsymbol{\Sigma}_{0,q}^{-1}(\boldsymbol{A}+\widetilde{\boldsymbol{A}})) + q^2\frac{\Delta t^2}{8}\left(\text{tr}^2(\boldsymbol{\Sigma}_{0,q}^{-1}(\boldsymbol{A}+\widetilde{\boldsymbol{A}})) - 2\text{tr}(\boldsymbol{\Sigma}_{0,q}^{-1}\boldsymbol{A}\boldsymbol{\Sigma}_{0,q}^{-1}\boldsymbol{A} + \text{tt.})\right)\right) \quad (84)$$

**Computation of** $\det \boldsymbol{J}$

Going back to $\det \boldsymbol{J}$, we are left with

$$\det \boldsymbol{J} = \left(1 + q\frac{\Delta t}{2}\text{tr}(\boldsymbol{\Sigma}_{0,q}^{-1}(\boldsymbol{A} + \tilde{\boldsymbol{A}})) + q^2\frac{\Delta t^2}{8}\left(\text{tr}^2(\boldsymbol{\Sigma}_{0,q}^{-1}(\boldsymbol{A} + \tilde{\boldsymbol{A}})) - 2\text{tr}(\boldsymbol{\Sigma}_{0,q}^{-1}\boldsymbol{A}\boldsymbol{\Sigma}_{0,q}^{-1}\boldsymbol{A} + \text{tt.})\right)\right) \tag{85}$$

$$\times \left(1 + \Delta t\,\text{tr}\boldsymbol{L} + \frac{\Delta t^2}{2}\left(\text{tr}^2\boldsymbol{L} - \text{tr}\boldsymbol{L}^2 + 2\text{tr}\boldsymbol{G}\right)\right). \tag{86}$$

Using the result for $\text{tr}\boldsymbol{L}$ we see that the terms in $\Delta t$ cancel, and the final expansion is of order $\Delta t^2$. As a result, after simplifications we are left with

$$\det \boldsymbol{J} = \left(1 - q^2\frac{\Delta t^2}{4}\text{tr}(\boldsymbol{\Sigma}_{0,q}^{-1}\boldsymbol{A}\boldsymbol{\Sigma}_{0,q}^{-1}\boldsymbol{A} + \text{tt.}) + \frac{\Delta t^2}{2}\left(-\text{tr}\boldsymbol{L}^2 + 2\text{tr}\boldsymbol{G}\right)\right). \tag{87}$$

Simplifying further we find that

$$\det \boldsymbol{J} = 1 + \frac{\Delta t^2 q}{2}\left(\frac{q}{2}\text{tr}(\boldsymbol{\Sigma}_{0,q}^{-1}(\boldsymbol{A} - \tilde{\boldsymbol{A}})\boldsymbol{\Sigma}_{0,q}^{-1}(\boldsymbol{A} - \tilde{\boldsymbol{A}}))\right. \tag{88}$$

$$\left. + \sum_{i,j}\frac{u_j(1 - u_iq)(1 - u_jq)(1 - q(u_j + u_i))}{(u_i + u_j)^2}\text{tr}\left(\boldsymbol{W}_i(\boldsymbol{A} - \tilde{\boldsymbol{A}})\boldsymbol{W}_j(\boldsymbol{A} - \tilde{\boldsymbol{A}})\right)\right) \tag{89}$$

This can be simplified

$$\left(\frac{q}{2}\text{tr}(\boldsymbol{\Sigma}_{0,q}^{-1}(\boldsymbol{A} - \tilde{\boldsymbol{A}})\boldsymbol{\Sigma}_{0,q}^{-1}(\boldsymbol{A} - \tilde{\boldsymbol{A}}))\right. \tag{90}$$

$$\left. + \sum_{i,j}\frac{u_j(1 - u_iq)(1 - u_jq)(1 - q(u_j + u_i))}{(u_i + u_j)^2}\text{tr}\left(\boldsymbol{W}_i(\boldsymbol{A} - \tilde{\boldsymbol{A}})\boldsymbol{W}_j(\boldsymbol{A} - \tilde{\boldsymbol{A}})\right)\right) \tag{91}$$

$$= \sum_{i,j}\frac{(1 - qu_i)(1 - qu_j)}{(u_i + u_j)^2}\left(\frac{q}{2}u_i^2 - \frac{q}{2}u_j^2 + u_j\right)\text{tr}\left(\boldsymbol{W}_i(\boldsymbol{A} - \tilde{\boldsymbol{A}})\boldsymbol{W}_j(\boldsymbol{A} - \tilde{\boldsymbol{A}})\right). \tag{92}$$

which leaves us with the following simplification

$$\det \boldsymbol{J} = 1 + \frac{\Delta t^2 q}{2}\sum_{i,j}\frac{u_i(1 - u_iq)(1 - u_jq)}{(u_i + u_j)^2}\text{tr}\left(\boldsymbol{W}_i(\boldsymbol{A} - \tilde{\boldsymbol{A}})\boldsymbol{W}_j(\boldsymbol{A} - \tilde{\boldsymbol{A}})\right)\right). \tag{93}$$

Using $\boldsymbol{\Sigma}_0 = \sum_i \sigma_i^2\boldsymbol{W}_i$, we can compute the eigenvalue decomposition as a function of $\sigma_i$. Using the notation $\sigma_{i,q}^2 = 1 + q\sigma_i^2$ we have

$$\frac{u_i(1 - u_iq)(1 - u_jq)}{(u_i + u_j)^2} = \frac{\sigma_i^2\sigma_{j,q}^2}{(\sigma_{j,q}^2\sigma_i^2 + \sigma_{i,q}^2\sigma_j^2)^2}. \tag{94}$$

Finally, at first non-zero order in $\Delta t$ the determinant reads

$$\det \boldsymbol{J} = 1 + \frac{\Delta t^2 q}{2}\sum_{i,j}\frac{\sigma_i^2\sigma_{j,q}^2}{(\sigma_{j,q}^2\sigma_i^2 + \sigma_{i,q}^2\sigma_j^2)^2}\text{tr}\left(\boldsymbol{W}_i(\boldsymbol{A} - \tilde{\boldsymbol{A}})\boldsymbol{W}_j(\boldsymbol{A} - \tilde{\boldsymbol{A}})\right). \tag{95}$$

**Taylor expansion of the GHK term**

We now expand the masses and mean at first order in $\Delta t$, $m_{\Delta t} = m_0 + \Delta t m_0 h$, $\tilde{m}_{\Delta t} = m_0 + \Delta t m_0\tilde{h}$, $\boldsymbol{\mu}_{\Delta t} = \boldsymbol{\mu}_0 + \Delta t\boldsymbol{v}$, $\tilde{\boldsymbol{\mu}}_{\Delta t} = \boldsymbol{\mu}_0 + \Delta t\tilde{\boldsymbol{v}}$. We have at first non zero order in $\Delta t$

$$\exp\left[-\frac{1}{2}(\boldsymbol{\mu}_{\Delta t} - \tilde{\boldsymbol{\mu}}_{\Delta t})^\top\boldsymbol{X}_{\Delta t}^{-1}(\boldsymbol{\mu}_{\Delta t} - \tilde{\boldsymbol{\mu}}_{\Delta t})\right] = 1 - \frac{\Delta t^2}{2}(\boldsymbol{v} - \tilde{\boldsymbol{v}})^\top\boldsymbol{X}_0^{-1}(\boldsymbol{v} - \tilde{\boldsymbol{v}}). \tag{96}$$

We denote $\boldsymbol{X}_0 = 2\boldsymbol{\Sigma}_0 + q^{-1}$. Expanding the remaining terms, the zeroth and first order terms cancel, leading to the expansion

$$\mathrm{GHK}\left(m_{\Delta t}\mathcal{N}(\boldsymbol{\mu}_{\Delta t}, \boldsymbol{\Sigma}_{\Delta t}), \widehat{m}_{\Delta t}\mathcal{N}(\widehat{\boldsymbol{\mu}}_{\Delta t}, \widehat{\boldsymbol{\Sigma}}_{\Delta t})\right) \tag{97}$$

$$= m_0 q^{-1}\Delta t^2 \left(\left((\boldsymbol{v} - \widetilde{\boldsymbol{v}})^\top \boldsymbol{X}_0^{-1}(\boldsymbol{v} - \widetilde{\boldsymbol{v}}) + \frac{1}{2}(h - \widetilde{h})^2\right)\right. \tag{98}$$

$$\left. + q\sum_{i,j}\frac{\sigma_i^2\sigma_{j,q}^2}{(\sigma_{j,q}^2\sigma_i^2 + \sigma_{i,q}^2\sigma_j^2)^2}\mathrm{tr}\left(\boldsymbol{W}_i(\boldsymbol{A} - \widetilde{\boldsymbol{A}})\boldsymbol{W}_j(\boldsymbol{A} - \widetilde{\boldsymbol{A}})\right)\right). \tag{99}$$

Integrating over all snapshots, the continuous time loss reads

$$\mathcal{L} = \int_0^T m_t q^{-1}\left(\left((\boldsymbol{v} - \widetilde{\boldsymbol{v}})^\top \boldsymbol{X}_0^{-1}(\boldsymbol{v} - \widetilde{\boldsymbol{v}}) + \frac{1}{2}(h - \widetilde{h})^2\right)\right.$$

$$\left. + q\sum_{i,j}\frac{\sigma_i^2\sigma_{j,q}^2}{(\sigma_{j,q}^2\sigma_i^2 + \sigma_{i,q}^2\sigma_j^2)^2}\mathrm{tr}\left(\boldsymbol{W}_i(\boldsymbol{A} - \widetilde{\boldsymbol{A}})\boldsymbol{W}_j(\boldsymbol{A} - \widetilde{\boldsymbol{A}})\right)\right)dt$$

$$+ \lambda\int_0^T \int \rho_t(\boldsymbol{x})\left(\|\boldsymbol{v}_t(\boldsymbol{x})\|_2^2 + \alpha|g_t(\boldsymbol{x})|^2\right)d\boldsymbol{x}dt. \tag{100}$$

The first order expansions $\boldsymbol{A}$, $h$, and $\boldsymbol{v}$ are obtained directly using the ODEs in Prop. 2.1, giving the final result.

Let's denote $\theta_t \in \mathbb{R}^N$ the vector of the $N$ parameters defining the growth and the drift at time $t$. We study the functional $\mathcal{L}[\theta_t]$ which reads

$$\mathcal{L}[\theta_t] = \int_0^T (F_t(\theta_t) + \lambda R_t(\theta_t))dt. \tag{101}$$

where $F$ is the part of the integrand coming from the expansion of the Gaussian-Hellinger-Kantorovich, and $R$ is the regularisation.

Let's take $t \in [0, T]$. Because $\theta \mapsto F_t(\theta)$ is the composition of convex functions and of affine maps, $\theta \mapsto F_t(\theta)$ is also convex. Let's show that $\theta \mapsto R_t(\theta)$ is a strongly convex function.

**Lemma B.1.** *Let $w : (\theta, x) \in \mathbb{R}^N \times \mathbb{R}^n \mapsto \mathbb{R}^d$ be a function continuous in $x$ and linear in $\theta$. Let $f : \mathbb{R}^d \mapsto \mathbb{R}$ be a continuous, strictly convex function and $\rho$ a continuous function from $\mathbb{R}^n$ to $]0, +\infty[$. We then have that*

$$\ell : \theta \in \mathbb{R}^N \mapsto \int \rho(x)f(w(\theta, x))dx \tag{102}$$

*is strictly convex.*

*Proof.* Let's take $\theta_1 \neq \theta_2$ and $\alpha \in ]0, 1[$. By linearity, we have $\forall x$:

$$w(\alpha\theta_1 + (1 - \alpha)\theta_2, x) = \alpha w(\theta_1, x) + (1 - \alpha)w(\theta_2, x). \tag{103}$$

Because $\theta \mapsto f(w(\theta, x))$ is a the composition of a convex function and of a linear map, it is convex for all $x$. Then, $\forall x$,

$$f(w(\alpha\theta_1 + (1-\alpha)\theta_2, x)) = f(\alpha w(\theta_1, x) + (1-\alpha)w(\theta_2, x)) \leqslant \alpha f(w(\theta_1, x)) + (1-\alpha)f(w(\theta_1, x)). \tag{104}$$

Because $w$ is linear in $\theta$, each of its component in $\mathbb{R}^d$ is a multivariate polynomial function of $\theta$ of degree one. By uniqueness of the coefficient of polynomial functions, because $\theta_1 \neq \theta_2$ there exists $x_0$ such that $w(\theta_1, x_0) \neq w(\theta_2, x_0)$. By continuity, this is also true on an open set $U \subset \mathbb{R}^n$ centred in $x_0$. Therefore, since $f$ is strictly convex, the inequality is strict for all $x \in U$, i.e.:

$$f(w(\alpha\theta_1 + (1-\alpha)\theta_2, x)) = f(\alpha w(\theta_1, x) + (1-\alpha)w(\theta_2, x)) < \alpha f(w(\theta, x)) + (1-\alpha)f(w(\theta, x)). \tag{105}$$

By multiplying by $\rho(x)$, which is strictly positive, and by integrating, we keep the inequality strict and we find

$$\ell(\alpha\theta_1 + (1 - \alpha)\theta_2) < \alpha\ell(\theta_1) + (1 - \alpha)\ell(\theta_2), \tag{106}$$

proving that $\ell$ is strictly convex. $\qquad\square$

|  | Hidden layers | Batch size | Learning rate | Iterations |
|---|---|---|---|---|
| Bistable ($d \leqslant 10$) | 64, 64, 64 | 256 | 0.003 | 25,000 |
| Bistable ($d > 10$) | 256, 256, 256 | 256 | 0.001 | 100,000 |
| Bifurcating CLE | 64, 64, 64 | 256 | 0.01 | 10,000 |
| Haematopoietic CLE | 128, 128, 128 | 256 | 0.01 | 10,000 |
| Lineage tracing | 128, 128, 128 | 256 | 0.01 | 10,000 |

Table 5: Hyperparameter settings: score networks

|  |  | Hidden (force) | Hidden (growth) | Batch | LR | Iterations | $\lambda$ | $\alpha$ | $\gamma$ |
|---|---|---|---|---|---|---|---|---|---|
| UPFI | Bistable ($d \leqslant 10$) | 64, 64, 64 | 64, 64, 64 | 256 | 0.003 | 5,000 | see details |  | 5 |
|  | Bistable ($d > 10$) | 256, 256, 256 | 256, 256, 256 | 256 | 0.001 | 25,000 | see details |  | 5 |
|  | Bif CLE | 64, 64, 64 | 64 | 256 | 0.003 | 10,000 | 0.001 | 1 | 5 |
|  | Haem CLE | 128, 128, 128 | 128 | 256 | 0.003 | 10,000 | 0.001 | 1 | 5 |
|  | Lineage | 128, 128, 128 | 128, 128, 128 | 256 | 0.001 | 10,000 | 0.001 | 1 | 25 |
| PFI | Bistable ($d \leqslant 10$) | 64, 64, 64 | – | 256 | 0.003 | 5,000 | see details |  | – |
|  | Bistable ($d > 10$) | 256, 256, 256 | – | 256 | 0.001 | 25,000 | see details |  | – |
|  | Bif CLE | 64, 64, 64 | – | 256 | 0.003 | 10,000 | 0.001 | – | – |
|  | Haem CLE | 128, 128, 128 | – | 256 | 0.003 | 10,000 | 0.001 | – | – |
|  | Lineage | 128, 128, 128 | – | 256 | 0.001 | 10,000 | 0.001 | – | – |
| ODE | Bistable ($d \leqslant 10$) | Coupled to growth | 64, 64, 64 | 256 | 0.003 | 5,000 | see details |  | 5 |
|  | Bistable ($d > 10$) | Coupled to growth | 256, 256, 256 | 256 | 0.001 | 25,000 | see details |  | 5 |
|  | Lineage | Coupled to growth | 128, 128, 128 | 256 | 0.001 | 10,000 | 0.001 | 0.1 | 25 |
| TIGON++ | Bistable ($d \leqslant 10$) | 64, 64, 64 | 64, 64, 64 | 256 | 0.003 | 5,000 | see details |  | 5 |
|  | Bistable ($d > 10$) | 256, 256, 256 | 256, 256, 256 | 256 | 0.001 | 25,000 | see details |  | 5 |
|  | Lineage tracing | 128, 128, 128 | 128, 128, 128 | 256 | 0.001 | 10,000 | 0.001 | 0.1 | 25 |

Table 6: Hyperparameter settings: dynamics. For bistable system certain hyperparameters chosen as a function of $d$, see Section C.2 for details.

We have $\rho_t(x) > 0$ for all $x$ because it is a Gaussian density. The drift and the fitness being linear in $\theta$, this shows that

$$\theta \mapsto R_t(\theta) = \int \rho_t(\boldsymbol{x}) \left( \|\boldsymbol{v}_t(\boldsymbol{x})\|_2^2 + \alpha |g_t(\boldsymbol{x})|^2 \right) d\boldsymbol{x} \tag{107}$$

is a strictly convex function of $\theta$. Additionally, it is a multivariate polynomial function of $\theta$ of degree two, so its Hessian is a definite positive and constant, proving that $\theta \mapsto R_t(\theta)$ is strongly convex, and so is $F_t + R_t$. Along with the fact that both $(t, \theta) \mapsto F_t(\theta)$ and $(t, \theta) \mapsto R_t(\theta)$ are continuous in $[0, T] \times \mathbb{R}^N$, this is enough to ensure the existence and uniqueness of a minimum $t \mapsto \theta_t^\star$ for the functional $\mathcal{L}[\theta_t]$ [51]. $\qquad\square$

# C    Implementation and experiment details

We implement Alg. 1 using PyTorch and employ the GeomLoss package [52] for computation of the unbalanced Sinkhorn divergence. All model training was carried out using a NVIDIA L40S GPU. Code is available at `https://github.com/zsteve/UPFI`.

## C.1    Score matching

While in principle any score matching approach to learn $\boldsymbol{s}_t(\boldsymbol{x}) = \nabla \log p_t(\boldsymbol{x})$ can be used within Alg. 1, in practice we employ denoising score matching [30] within the noise-conditional score network framework introduced in [5]. For $K + 1$ snapshots taken at times $(t_i)_{i=0}^K$, we parameterise the time-dependent score using a multilayer perceptron (MLP) $\boldsymbol{s}_\phi(t, \boldsymbol{x}, \eta) = \mathrm{NN}_\phi(d + 2, d)$ where $d$ is the dimension and $\eta$ is the noise level, and train using the algorithm described in [5, Section 4.2]. While a range of noise levels $\eta_0 < \ldots < \eta_L$ are used for training the score, subsequently for training the probability flow we use the smallest noise scale $\eta = \eta_0$, representing our final estimate of the score.

In what follows, we use a default of $L = 5$ noise levels logarithmically spaced between $(\exp(-2), 1)$. Score networks $\boldsymbol{s}_t(\boldsymbol{x})$ are parameterised using MLPs with ReLU activations. In all case, score training was carried out using noise-conditional denoising score matching [5, 30] using the AdamW optimiser with the hyperparameter choices listed in Table 5 .

## C.2 Training: UPFI and PFI

**Additive noise models** For UPFI, in all cases we parameterise an *autonomous* force $\boldsymbol{v}_\theta(\boldsymbol{x})$ using a MLP. This is because, in all simulated systems, the true force is also autonomous. In the experimental lineage tracing dataset we reason that an autonomous force would be consistent with the biologically motivated model of a Waddington's landscape [34]. We parameterise separately the force $\boldsymbol{v}_\theta(\boldsymbol{x})$ and growth $g_\theta(\boldsymbol{x})$ using MLPs with ReLU activations. We train UPFI models using the AdamW optimiser, our architecture and hyperparameter choices are listed in Table 6.

For PFI, motivated by the observations in the Gaussian case we reason that an autonomous force is insufficient to fit the data if growth is not accounted for. We therefore employ a *non-autonomous* force, parameterising $\boldsymbol{v}_\theta(t, \boldsymbol{x}) = \text{NN}_\theta(d+1, d)(t, \boldsymbol{x})$ with ReLU activations. We train PFI models using the AdamW optimiser, our architecture and hyperparameter choices are listed in Table 6.

**Multiplicative noise model** For the multiplicative noise models we considered in Fig. 5, we parameterise an autonomous force in two components, corresponding to production and degradation terms in the model (16). Specifically, we let

$$\boldsymbol{f}_\theta(\boldsymbol{x}) = \text{NN}_\theta(d, d)(\boldsymbol{x}), \quad \boldsymbol{g}_\theta(\boldsymbol{x}) = \text{NN}_\theta(d, d)(\boldsymbol{x}),$$

and the resulting force is $(\boldsymbol{f}_\theta - \boldsymbol{g}_\theta)(\boldsymbol{x})$. To constrain the output of both networks to be non-negative, we opt for a Softplus activation on the final layer of outputs. For all other layers we use the ReLU activation as a default choice. Architectures for $(\boldsymbol{f}, \boldsymbol{g})$ and all other hyperparameter choices are as given in Table 6.

**Scaling regularisation with dimension for bifurcating system** For the bifurcating system of Figure 3 and Table 1, it is necessary to scale the regularisation parameters $(\lambda, \alpha)$ as $d$ ranges in $\{2, 5, 10, 25, 50\}$. We reason that $\|\boldsymbol{v}\|^2 = \sum_{i=1}^d v_i^2$ grows roughly linearly with increasing dimension $d$, while the growth rate $|g|$ does not depend on $d$ since it is a scalar-valued field. This motivates the rescaled regularisation term

$$\lambda(d^{-1}\|\boldsymbol{v}\|^2 + \alpha|g|^2)$$

In practice we use $\lambda = 0.001$ and $\alpha = 0.1$ for all the values of $d$ considered.

Additionally, we alter aspects of training such as network size, learning rate, and number of training iterations depending on $d$. For $d > 10$ we used a wider network, paired with a smaller learning rate and more training iterations. These choices were consistently applied to all methods under consideration except for DeepRUOT due to difficulties with modifying the implementation.

## C.3 Training: fitness-ODE

Motivated by issues pertaining to the identifiability of dynamics involving both drift and growth, we propose a well-known dynamical model as a baseline model for inference. Let $\{\rho_t\}_t$ be a continuous distributional path satisfying mild conditions in the space of measures describing some population evolution. Then there exists a unique scalar field $U_t(\boldsymbol{x}_t)$ such that $\rho_t$ satisfies

$$\partial_t \rho_t(\boldsymbol{x}) = -\nabla \cdot (\rho_t(\boldsymbol{x})\nabla U_t(\boldsymbol{x})) + U_t(\boldsymbol{x})\rho(\boldsymbol{x}). \tag{108}$$

That this is the case can be read from [25, Section A.3] or [53, Proposition 2.2]. This can be interpreted as a continuous dynamics where $U_t(\boldsymbol{x})$ is the *fitness* of state $\boldsymbol{x}$. The rate at which agents reproduce is prescribed by $U_t(\boldsymbol{x})$, and agents migrate to regions of higher fitness following $\nabla U_t(\boldsymbol{x})$. Theoretically, for a regular enough sequence of population snapshots $\{\rho_t\}_t$, a single time-dependent fitness function $U_t$ is sufficient to generate the path $t \mapsto \rho_t$ via these dynamics. While it is perhaps not obvious, a *single* quantity, the fitness $U_t$, is enough to generate the full path $\rho_t$ in the space of measures. As a baseline, we therefore propose a neural parameterisation of $U_t$ and to learn $U_t$ following the TIGON++ setup, but with $\boldsymbol{v}_t(\boldsymbol{x}) = \nabla U_t(\boldsymbol{x})$ and $g_t(\boldsymbol{x}) = U_t(\boldsymbol{x})$. We parameterise $U_t(\boldsymbol{x}) = \text{NN}_\theta(d+1, 1)(t, \boldsymbol{x})$ using ReLU activations. All hyperparameter choices are listed in Table 6.

## C.4 Training: TIGON++

The problem of dynamical transport for systems with mass imbalance was previously studied in the work [24]. In this work, the authors consider *deterministic* systems only and allow both the force

$\boldsymbol{v}_t(\boldsymbol{x})$ and growth $g_t(\boldsymbol{x})$ to be time dependent. However, the TIGON algorithm relies on kernel density estimation (KDE) from the input data [24, Methods] which is sensitive to the choice of kernel bandwidth and suffers from the curse of dimensionality as the dimension increases. The choice of data-fitting loss, in the form of minimising squared discrepancies between the predicted and KDE densities, adds to these difficulties: this loss relies point-wise on estimated densities and is thus not "geometry-aware" in the sense of optimal transport based losses [52]. Finally, propagating densities under flow models e.g. (4) are well known to be computationally costly. For these reasons, training the TIGON algorithm was infeasible for most of our numerical experiments

Because we needed a robust comparison baseline, we decided to implement the same model used by TIGON (deterministic transport with growth), but train it with the UPFI training procedure. With UPFI, we circumvent the need for density estimation by using the probability flow formulation of the Fokker-Planck equation. Together with the use of the unbalanced Sinkhorn divergence as the data fitting loss, we believe that this substantially improves the TIGON method and also makes for a more rigorous baseline to test against UPFI. We call TIGON++ this re-implementation of TIGON.

Specifically, we parameterise a drift $\boldsymbol{v}_\theta(t, \boldsymbol{x}) = \mathrm{NN}_\theta(d+1, d)(t, \boldsymbol{x})$ and growth $g_\theta(t, \boldsymbol{x}) = \mathrm{NN}_\theta(d+1, 1)(t, \boldsymbol{x})$ with ReLU activations. For a sampled data point $(\boldsymbol{x}_0, m_0 = 1)$ at $t = 0$, its state $(\boldsymbol{x}_{t_i}, m_{t_i})$ at each timepoint $t_i$ is simulated by forward integration of the system

$$\dot{\boldsymbol{x}}_t = \boldsymbol{v}_\theta(t, \boldsymbol{x}_t), \qquad \dot{m}_t = g_\theta(t, \boldsymbol{x}_t) m_t, \tag{109}$$

and we form the empirical distribution $\widehat{\rho}_{t_i}(\boldsymbol{x}) = \sum_{k=1}^{N_i} m_{k,t_i} \delta(\widehat{\boldsymbol{x}}_{k,t_i} - \boldsymbol{x})$ for each timepoint $t_i$. For the rest of the training procedure we use the same data-fitting loss as UPFI, i.e. (12).

## C.5  DeepRUOT

We use the existing DeepRUOT implementation provided by [27], which parameterises the force, growth rate and score function. Different to our method, however, the individual components of the dynamics are coupled via a physics-informed neural network (PINN)-type loss ([27, Section 5.3]) that aims to incorporate information from the governing Fokker-Planck equation. The training procedure DeepRUOT consists of multiple stages and involves first neural ODE training, followed by flow matching, and again neural ODE training. For the bistable system example (Fig. 3) we use their PyTorch implementation of [27, Algorithm 1]. For each of the drift, growth and score networks, three hidden layers of size 128 were used. For further details on DeepRUOT implementation and training we refer the reader to [27] and accompanying code.

## C.6  Flow matching

We consider optimal transport-conditioned flow matching (OTFM), a class of *simulation-free* methods for training flows that approximate dynamical OT [38]. Compared to the other methods, OTFM training does not require numerical integration during training. This comes at the cost of requiring the transport plan $\pi_{i,i+1}$, or coupling, between successive snapshots $(\rho_{t_i}, \rho_{t_{i+1}})$ to be computed in advance of training the flow model. Given a pair of distributions $(\rho, \rho')$ with a OT coupling $\pi$ on $t \in [0, 1]$, OTFM trains a flow network $\boldsymbol{v}_\theta(t, \boldsymbol{x})$ by minimising a nonlinear least squares objective:

$$\min_\theta \ \mathbb{E}_{t \in U[0,1]} \ \mathbb{E}_{(\boldsymbol{x}, \boldsymbol{x}') \sim \pi} \|(\boldsymbol{x}' - \boldsymbol{x}) - \boldsymbol{v}_\theta(t, (1-t)\boldsymbol{x} + t\boldsymbol{x}')\|_2^2, \tag{110}$$

where $\pi$ is the optimal coupling obtained by solving the entropic OT problem:

$$\min_{\pi : \pi \mathbf{1} = \rho, \pi^\top \mathbf{1} = \rho'} \frac{1}{2} \mathbb{E}_{(x,x') \sim \pi} \|x - x'\|_2^2 + \varepsilon \mathrm{KL}(\pi | \rho \otimes \rho'). \tag{111}$$

We refer the reader to [38] for an in-depth discussion. We implement OTFM for dynamics inference by applying (110) across each pair of snapshots $(t_i, t_{i+1})$ to learn a single network $\boldsymbol{v}_\theta(t, \boldsymbol{x})$. To generate results shown in Table 1 we compute entropic OT couplings between pairs of snapshots using a squared Euclidean cost and pick $\varepsilon = \sigma^2(t_{i+1} - t_i)$.

By default, OTFM treats the balanced case of transport and is not appropriate for modelling dynamics with growth. As an additional baseline, we try an unbalanced variant, UOTFM, where the coupling $\pi$ is obtained by solving the *unbalanced* relaxation [23] of (111), where the hard marginal constraint is replaced with a soft penalty with weight $\lambda > 0$:

$$\min_\pi \frac{1}{2} \mathbb{E}_{(x,x') \sim \pi} \|x - x'\|_2^2 + \varepsilon \mathrm{KL}(\pi | \rho \otimes \rho') + \lambda \left( \mathrm{KL}(\pi \mathbf{1} | \rho) + \mathrm{KL}(\pi^\top \mathbf{1} | \rho') \right). \tag{112}$$

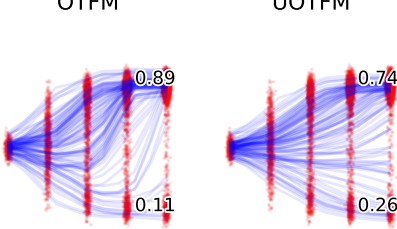

Figure 6: Sampled trajectories for trained flow matching models for bifurcating example, $d = 10$ (see Figure 4).

In practice we take $\lambda = N^{-2} \sum_{ij} \|x_i - x'_j\|^2$, i.e. the mean of the cost matrix. This is essentially the approach advocated for in [26]. In Figure 6 we show sampled trajectories from trained OTFM and UOTFM models for the bifurcating system of Section 3. As expected, OTFM results in trajectories biased towards the upper branch, while this is partially mitigated using UOTFM.

### C.7 Forward simulation

**Bistable system**  We consider a potential-driven dynamics in dimension $d \in \{2, 5, 10\}$ specified by

$$\boldsymbol{v} = -\nabla V, \qquad V(\boldsymbol{x}) = 0.9 \|\boldsymbol{x} - \boldsymbol{a}\|_2^2 \|\boldsymbol{x} - \boldsymbol{b}\|_2^2 + 10 \sum_{i=3}^{d} x_i^2$$
$$b(\boldsymbol{x}) = \tfrac{5}{2}(1 + \tanh(2x_0)),$$
$$d(\boldsymbol{x}) = 0.$$

For $t \in [0, 1]$ we simulate particles following $\mathrm{d}\boldsymbol{X}_t = \boldsymbol{v}(\boldsymbol{X}_t)\,\mathrm{d}t + \sigma\,\mathrm{d}\boldsymbol{B}_t$ using the Euler-Maruyama method. We set the noise level to $\sigma = 1/2$, and use the initial condition $\boldsymbol{X}_0 \sim \mathcal{N}(\boldsymbol{0}, 0.01\boldsymbol{I})$. At each Euler step, simulated particles divide with probability $b(\boldsymbol{X})\Delta t$. We simulate starting from $N_0 = 500$ particles, and the total population size grows over time following the prescribed dynamics. Population snapshots are taken from *independent* realisations of the process at $K + 1 = 5$ timepoints uniformly spaced between $[0, 1]$.

**Reaction network systems**  The bifurcating and HSC reaction networks were taken from previous literature [40, 41], corresponding to the networks BF and HSC in the collection of BoolODE benchmarking problems [40]. The original implementation, however, modelled both gene and protein expression levels and as a result does not strictly fall in the modelling framework we consider. This is because protein levels are not observed and thus are hidden variables. We re-implemented each of these systems to involve only gene expression dynamics, and also change the noise model: in the original implementation an ad-hoc square-root noise model was used, i.e. $\boldsymbol{\sigma}(\boldsymbol{x}) = \alpha\sqrt{\boldsymbol{x}}$. We choose to use a more biophysically motivated noise model (16), and take

$$\sigma_i(\boldsymbol{x}) = \sqrt{f_i(\boldsymbol{x}) + \lambda_i x_i},$$

where $\boldsymbol{f}(\boldsymbol{x})$ is a vector-valued function of state-dependent production rates for each gene $x_i$, and each $\lambda_i$ is the corresponding degradation rate. For the growth rates, we consider a scenario where cells in one branch of the system trajectory divide at a faster rate than the others:

- In the bifurcating network, we set

$$\beta(x) = 5 \left( \frac{\tanh(5(x_7 - 0.7)) + 1}{2} \right) \left( 1 - \frac{\tanh(5(x_1 - 0.7)) + 1}{2} \right)$$

- In the HSC network, we set

$$\beta(x) = 5.5 \left( \frac{\tanh(x_{\mathrm{E}} - 1.5) + 1}{2} \right).$$

We simulate both systems using the same code as for the bistable system, with the volume parameter $1/\sqrt{V} = 0.5$, starting with a population of 500 cells and capturing $K + 1 = 10$ timepoints. For bifurcating and HSC network we use simulation time intervals of $0 \leqslant t \leqslant 1.25$ and $0 \leqslant t \leqslant 1$ respectively.

**Fate probability computation**   We provide a straightforward definition of fate probability in what follows. Let $\boldsymbol{x}_t$ be the state of an observed cell or individual at time $t$. Let $\sqcup_i \Omega_i$ be a partitioning of the state space (e.g. some subset of $\mathbb{R}^d$) where each $\Omega_i$ is understood to correspond to well-defined, stable states of the system at some final time, say $t = 1$. In the biological setting, this is typically thought of as a mature cell "type" [1, 34]. Then the fate probability of $\boldsymbol{x}_t$ towards $\Omega_i$ is defined as the conditional probability:
$$\mathbb{P}(\boldsymbol{X}_1 \in \Omega_i | \boldsymbol{X}_t = \boldsymbol{x}_t).$$
In practice, such as for the bistable system of Fig. 3, we form a partitioning of the state space into two regions by running $k$-means with $k = 2$ on the final snapshot from the system. Given any query state $\boldsymbol{x}_t$ at time $t$, we empirically estimated true and inferred fate probabilities by forward simulation of either the ground truth SDE or inferred dynamics:

$$\mathbb{P}(\boldsymbol{X}_1 \in \Omega_i | \boldsymbol{X}_t = \boldsymbol{x}_t) \approx M^{-1} \sum_{i=1}^{M} \mathbf{1}_{\Omega_i}(\boldsymbol{X}_1 | \boldsymbol{X}_t = \boldsymbol{x}_t),$$

where $M$ is the number of trials to sample.

## C.8   Neural graphical model

For the Neural Graphical Model (NGM) example of Fig. 4, we use the architecture introduced in [42] as a drop-in parameterisation of the autonomous force $\boldsymbol{v}_\theta(\boldsymbol{x})$. For each output variable, we use two hidden layers with sizes $[64, 64]$. We use a group lasso regularisation strength $\lambda_{\mathrm{GL}} = 0.03$ and employ the proximal update scheme outlined in [42, Section C.4.1] with a learning rate of $0.003$ and train for $5,000$ iterations. We use the score networks that were already pre-trained for the additive and multiplicative UPFI models. All other training details are taken to be the same as for the earlier UPFI training.

## C.9   Single cell lineage tracing data

**Preprocessing**   Data for the study of [1] are available from the original publication using the GEO database with accession number `GSE140802`. Starting from raw counts, expression data is normalised using `dyn.pp.recipe_monocle` function from the Dynamo package [43]. In brief, raw gene expression values are per-cell normalised and then $\log(1 + x)$-transformed. For all our experiments we use the 10-dimensional PCA embedding of cell gene expression profiles. Spliced and unspliced transcript counts were obtained from reanalysis of the raw sequencing data of [1] and RNA velocity estimates were subsequently obtained using the Dynamo package [43]. Scripts and datasets for this re-analysis are available upon request.

From the full dataset, 86,416 cells deemed to be contributing to the "Neutrophil-Monocyte" trajectory (as determined by the original publication [1]) were selected. Using the 10-dimensional PCA embedding for these data, we apply UPFI, PFI, fitness-ODE and TIGON. We do not include Deep-RUOT in this analysis since it resulted in an out-of-memory error in the initial stages of training. Noting also that the original publication [27] considered only the 2D SPRING layout, be believe that further modification of its training pipeline may be necessary.

**Training**   For UPFI, we train a time-dependent score model with hidden dimensions $[128, 128, 128]$ for $10,000$ iterations with a batch size of 256 and learning rate of $10^{-2}$. We adopt an additive noise model and parameterise an autonomous force $\boldsymbol{v}_\theta(\boldsymbol{x})$ and growth $g_\theta(\boldsymbol{x})$ each with a MLP with hidden dimensions $[128, 128, 128]$. We set $\gamma = 25.0, \lambda = 0.001, \alpha = 1.0$ and we set $\sigma = 0.5$. Note that the choice of $\gamma$ is not scale-invariant, and we found that the typical length scale in the lineage tracing data is larger than in the simulation data. We train for $10,000$ iterations with a batch size of 256 and learning rate $10^{-3}$.

We train PFI with the same hyperparameter choices as UPFI, except we use a non-autonomous force as done in earlier examples. Finally, we train TIGON and fitness-ODE following the training procedure outlined in Sections C.4 and C.3 and the same hyperparameters as for UPFI and PFI.

| $d$ | Score matching | UPFI | PFI | ODE | TIGON |
|---|---|---|---|---|---|
| 25 | $249.73 \pm 0.50$ | $439.48 \pm 1.51$ | $333.51 \pm 2.02$ | $861.41 \pm 11.43$ | $313.79 \pm 3.71$ |
| 50 | $250.71 \pm 1.52$ | $443.52 \pm 2.23$ | $335.94 \pm 3.65$ | $868.93 \pm 11.86$ | $316.18 \pm 4.17$ |

| $d$ | OTFM | UOTFM |
|---|---|---|
| 25 | $100.68 \pm 0.84$ | $52.64 \pm 0.91$ |
| 50 | $99.61 \pm 0.68$ | $52.95 \pm 0.75$ |

Table 7: Runtimes (s) for the different methods on the bistable system, for $d = 25, 50$.

| **Cosine similarity force field** | | | | **Growth rate correlation** | | |
|---|---|---|---|---|---|---|
| | $\lambda = 0.0$ | $0.01$ | $0.1$ | $0.0$ | $0.01$ | $0.1$ |
| $\alpha = 0$ | 0.105 | 0.059 | 0.078 | 0.923 | 0.917 | 0.917 |
| 0.01 | 0.105 | 0.059 | 0.060 | 0.923 | 0.917 | 0.899 |
| 0.1 | 0.105 | 0.086 | 0.203 | 0.923 | 0.909 | 0.349 |

Table 8: Results for regularisation ablation experiments for the bistable system with $d = 10$.

### C.10 Remark on UPFI in the case when only frequencies are available

When $N_i/N_0$ is not a good estimator for the ratio $|\rho_{t_i}|/|\rho_0|$, we can only build an estimator for the normalised density $\rho_{t_i}/|\rho_{t_i}|$:

$$\frac{\rho_{t_i}}{|\rho_{t_i}|} \simeq \frac{1}{N_i} \sum_{k=1}^{N_i} \delta(\boldsymbol{x} - \boldsymbol{x}_{k,t_i}) \tag{113}$$

This poses limitations on the fitness which can be inferred. Indeed, writing $\widetilde{\rho}_t = \rho_t/|\rho_t|$, we have

$$\partial_t \widetilde{\rho}_t = \partial_t \left( \frac{\rho_t}{|\rho_t|} \right) = \frac{1}{|\rho_t|} \partial_t \rho_t - (\partial_t \log |\rho_t|) \widetilde{\rho}_t.$$

Substituting back the PDE governing $\rho_t$, we find that $\widetilde{\rho}_t$ is also governed by a drift-diffusion PDE, but with a time-dependent bias in the source term:

$$\partial_t \widetilde{\rho}_t = -\boldsymbol{\nabla} \cdot \left[ \widetilde{\rho}_t \left( \boldsymbol{v}_t - \boldsymbol{\nabla} \cdot \boldsymbol{D}_t - \boldsymbol{D}_t \boldsymbol{\nabla} \log \widetilde{\rho}_t \right) \right] + (g_t - \partial_t \log |\rho_t|) \widetilde{\rho}_t.$$

Therefore, when we don't have access to the absolute number of individuals present in a population at a given time, we can only hope to infer the fitness $g_t$ up to a time-dependent bias. In this case, the UPFI approach can also be applied using a large enough mass conservation strength $q$ in the unbalanced Sinkhorn distance.

### C.11 Ablation experiments

We performed ablation experiments for the regularisation on the bifurcating system with $d = 10$. In Table 8, we show for varying $\alpha$ and $\lambda$ the cosine error for force recovery as well as Pearson correlation for growth rate recovery. When $\lambda = 0$, there is no regularisation in the loss function, and the resulting error is larger than for $\lambda > 0$. From the growth rate perspective, however, the unregularised model performs better, albeit at the cost of higher force field error.

These results suggest that the growth rate is less sensitive to regularisation, which probably stems from the implicit regularisation arising from the relatively small neural network sizes. Since the additional loss terms are included only to ensure uniqueness, as motivated by the theorem in the main text, this ablation study suggests a simple rule of thumb for UPFI: use little or no regularisation when employing moderately sized neural networks.

