# OpenReview forum: "Inferring stochastic dynamics with growth from cross-sectional data"
_NeurIPS.cc/2025/Conference — NeurIPS 2025 poster_

### Official Review · Reviewer_rQy3 · 2025-06-10

**Clarity:** 4
**Significance:** 1
**Originality:** 3
**Rating:** 4
**Confidence:** 3

**Summary:**

The paper proposes **Unbalanced Probability Flow Inference (UPFI)**, an extension of Probability-Flow Inference that incorporates a **growth (birth-death) term** into the Lagrangian formulation of the Fokker–Planck equation. UPFI proceeds in two steps:
1. **Score estimation** of ∇log ρ_t via denoising score matching -- using denoising score matching to estimate the time-dependent score of the data distribution
2. **Neural-ODE fitting** of drift *v*(x,t) and growth *g*(x,t) by pushing weighted particles through the characteristic ODE and minimizing an **unbalanced Sinkhorn divergence** plus a Wasserstein–Fisher–Rao (WFR) regularizer -- learning the drift and growth terms by solving a characteristic ODE system.

The authors provide a theoretical analysis, demonstrating the non-identifiability between drift and growth. The method is evaluated on a series of simulated systems, including a high-dimensional bistable system and chemical Langevin equation models, as well as on a real-world single-cell RNA-seq dataset of monocyte-neutrophil development.

**Questions:**

1. **Positioning** - How does the framework position itself relative to the recent literature on flow matching [1-6]? These methods have been extended to stochastic and unbalanced settings. A discussion of the relative merits would strengthen the paper.
2. **Compute budget** - Could the authors provide a more detailed analysis of the computational complexity and scalability of UPFI? Specifically, how does the training time scale with the number of samples, dimensions, and the number of time points?
3. **Velocity prior:** Could kernel-smoothed RNA-velocity guide or evaluate the drift, as in [4]?
4. **Regulariser sensitivity:** How do drift and growth fields vary with different λ and α values? Please provide an ablation.

**Ethical Concerns:**

["NO or VERY MINOR ethics concerns only"]

**Final Justification:**

The rebuttal addressed scalability concerns as well as provided additional context in terms of other methods.

**Limitations:**

The authors have adequately addressed the primary theoretical limitation of their work: the non-identifiability of drift and growth. The significant computational cost and scalability issues inherent in their simulation-based training scheme is not discussed.

**Paper Formatting Concerns:**

No concerns

**Quality:**

2

**Strengths And Weaknesses:**

#  Strengths
* **Problem Relevance** - tackles a crucial and challenging problem in computational biology: learning cell dynamics from static snapshots, accounting for population growth and death.
* **Identifiability analysis** - provides an analytic example (linear OU + quadratic fitness) and proves uniqueness under WFR. This provides a solid theoretical foundation for the necessity of WFR regularization.
* **Unbalanced OT loss** - handles unequal cell counts directly, an important practical feature for real lineage data.
* **Clarity** -  well-written, clearly structured, and easy to follow


# Weaknesses

* **Limited scalability** -  relies on an ODE solver within the main training loop (Algorithm 1) raises questions about the method's computational scalability. While the method is presented as a general solution, the experiments are restricted to low-dimensional PCA embeddings (d=2, 5, 10 for simulations; d=10 for real data).
* **Contribution feels narrow** - While the problem is important, the proposed solution feels like a very incremental improvement. Given the severe scalability concerns and the lack of engagement with state-of-the-art generative modeling techniques (like flow matching), the paper's impact seems limited. The recent work [1-6] suggests that the field is moving towards more scalable solutions, questioning the long-term relevance of the proposed method.
* **Related-work gap** - paper does not discuss the advancing field of flow matching [1-5], while it is far more relevant to the paper's goal of learning vector fields between distributions than some of the older cited works
* **Limited Ablations** - paper does not provide ablation along regularisation, which makes it difficult to understand how much hyper tuning is required given method is already expensive


[1] Tong, A., Fatras, K., Malkin, N., Huguet, G., Zhang, Y., Rector-Brooks, J., Wolf, G. and Bengio, Y., 2023. Improving and generalizing flow-based generative models with minibatch optimal transport. arXiv preprint arXiv:2302.00482.

[2] Tong A, Malkin N, Fatras K, Atanackovic L, Zhang Y, Huguet G, Wolf G, Bengio Y. Simulation-free schr\" odinger bridges via score and flow matching. arXiv preprint arXiv:2307.03672. 2023 Jul 7.

[3] Kapusniak, K., Potaptchik, P., Reu, T., Zhang, L., Tong, A., Bronstein, M., Bose, J. and Di Giovanni, F., 2024. Metric flow matching for smooth interpolations on the data manifold. Advances in Neural Information Processing Systems, 37, pp.135011-135042.

[4] Petrović, K., Atanackovic, L., Kapuśniak, K., Bronstein, M.M., Bose, J. and Tong, A., 2025, March. Curly flow matching for learning non-gradient field dynamics. In Learning Meaningful Representations of Life (LMRL) Workshop at ICLR 2025.

[5] Eyring L, Klein D, Uscidda T, Palla G, Kilbertus N, Akata Z, Theis F. Unbalancedness in neural monge maps improves unpaired domain translation. arXiv preprint arXiv:2311.15100. 2023 Nov 25.

[6] Wang, D., Jiang, Y., Zhang, Z., Gu, X., Zhou, P. and Sun, J., 2025. Joint Velocity-Growth Flow Matching for Single-Cell Dynamics Modeling. arXiv preprint arXiv:2505.13413.

---

> ### Author Rebuttal · Authors · 2025-07-31
>
> We thank the reviewer for their thorough review and appreciation of our efforts to provide a clear and theoretically justified method. Below we address comments about the i) relevance of our contribution and its positioning in the literature, ii) scalability of our algorithm, and iii) ablations.
>
> ### Contribution and positioning
> > Contribution feels narrow - While the problem is important, the proposed solution feels like a very incremental improvement. Given the severe scalability concerns and the lack of engagement with state-of-the-art generative modeling techniques (like flow matching), the paper's impact seems limited.
> > Positioning - How does the framework position itself relative to the recent literature on flow matching [1-6]?
>
> We would like to point out that [6] **became available online May 19 2025** and represents a concurrent work of which we **could not have been aware** at the time of our submission.
>
> To better position UPFI and its limitations in the literature, we would like to stress one important point: UPFI is a principled approach to learn, from cross-sectional samples, Fokker-Planck equations **with growth and arbitrary diffusion tensor**.
>
> As such, UPFI can fit not only processes with **additive noise** $\sigma=\mathrm{const}$ (as is typical of OT-based methods), but also models with state and drift-dependent **multiplicative** diffusion $\sigma(t, x)$. The relevance of such models in biological systems is well known [Coomer2022].
>
> To our knowledge, such models are are yet out of reach of any Flow Matching (FM) approach currently available, and UPFI is the first approach able to simultaneously account for growth and arbitrary, possibly multiplicative, intrinsic noise. This aspect of our work is illustrated on the simulated HSC example in Fig.4 and Table 2, for which fitting with additive noise ensures good interpolation, but does not extract correct regulatory interactions. We believe that this versatility of UPFI explains apparent limitations with respect to more recent FM methods.
>
> On the other hand, while FM enables simulation-free interpolants between distributions, these approaches sacrifice flexibility for scalability. A crucial ingredient in any FM approach is the choice of **coupling** between distributions, since this prescribes the dynamics. For **balanced** transport, Sinkhorn couplings are a reasonable choice thanks to Benamou-Brénier and reciprocal property of the Schrodinger bridge [Tong2023].
>
> On the other hand, we are not aware of theoretical arguments for a **good** choice of couplings in the unbalanced case, that have been applied to FM. [Eyring2023] consider using unbalanced OT with soft marginal constraints, under which the dynamics occur in two distinct steps: (i) a "reweighting" procedure involving inflation/deflation of mass density, and (ii) transport under the FM framework. Effectively, this **isolates** the effect of transport and growth. In the biological setting, and for general Fokker-Planck equations with source terms, however, transport and growth occur **simultaneously** and can interact. UPFI allows for these to be jointly modeled.
>
> For a rigorous extension of FM to the inference of diffusions with arbitrary intrinsic noise and growth, rigorous mathematical insight on the construction of coupling and choice of interpolant is still lacking.
> Thus, while we agree that UPFI lacks the scalability of FM, it can model systems of biological and physical relevance for which principled, theoretically justified FM approaches are unavailable at present.
>
> Moving towards more scalable solutions is indeed an important direction and we believe our paper provides a useful solid theoretical foundation for later works.
>
> **Thanks to this feedback, we now add a comparison to OT flow matching and an unbalanced variant [Eyring2023], see below and our response to other reviewers. We will expand discussion of related work, and emphasize what sets UPFI apart from FM methods.**
>
> ### Scalability
> > Limited scalability - relies on an ODE solver within the main training loop (Algorithm 1) raises questions about the method's computational scalability.
> > Compute budget - Could the authors provide a more detailed analysis of the computational complexity and scalability of UPFI? Specifically, how does the training time scale with the number of samples, dimensions, and the number of time points?
>
> To contextualize the scalability issue, we first want to point out that our method **already improves over existing methods like DeepRUOT and TIGON**, which applied their approach on the Lineage Tracing dataset in a **two-dimensional force layout embedding**.
> An improvement of our method is that, as opposed to TIGON and DeepRUOT, UPFI does not perform density estimation, although
> we agree with the reviewer that UPFI cannot scale to thousands of dimensions.
>
> Regarding the scalability and computational complexity of UPFI, we consider each step: i) score estimation, ii) sinkhorn loss and iii) ODE fitting. See also our response to `wcim`.
>
> Regarding the denoising score matching, its complexity is $\mathcal{O}(B\times d)$ where $d$ is the dimension, B is the batch size. For large entropic regularization, the unbalanced Sinkhorn divergence can be computed efficiently with a $\mathcal{O}(B^2)$ cost and with a dimension-independent sample complexity $\mathcal{O}(B^{-1/2})$ [Sejourne2019], provided $B$ is large enough for these scalings to old. We agree with the reviewer's concerns regarding the scalability of the ODE integration step, which in theory can limit the inference of dynamical models in high-dimensions. In practice, we mitigate this issue by taking ~2 to 3 Euler steps in between each time points. Overall, while our approach can't scale to hundreds or thousands of dimensions, we can reach moderately high dimensions provided we increase the size of the neural networks. We performed additional experiments with ~50 dimensions of the bifurcating example to showcase the ability of UPFI to scale to these moderately high-dimensions.
>
> We also compared UPFI with (unbalanced) flow matching and unbalanced flow matching (U)OTFM. While unbalanced OTFM offers indeed a fair improvement upon OTFM, we find that UPFI is consistently more accurate for vector field estimation and fate probabilities.
>
> **See table in our response to `wcim`: _Bifurcating ex. (Table 1) to $d = 25, 50$_**
>
> To give an idea of the runtimes of UPFI, we time for our method associated to the experiments of Table 1 and Fig 5, which we report below:
>
> ### Runtime (in seconds, for Table 1)
>
> |dim|Score|UPFI|DeepRUOT|
> |-:|:-|:-|-:|
> |2|61.32±0.50|86.83±0.62|1156.21|
> |5|61.21±0.43|86.99±0.38|1036.71|
> |10|61.26±0.42|87.22±0.51|1002.07|
> |25`new!`|249.54±1.00|442.58±2.01|1194.7|
> |50`new!`|249.25±0.34|440.84±1.87|1079.56|
>
> Note we add results for $d=25,50$, for which we increase score matching iters from 25k to 100k, and UPFI iters from 5k to 25k.
>
> ### Runtime (in seconds, for Fig.5)
>
> |Score|UPFI|
> |-:|-:|
> |24.19±0.13|207.97±1.63|
>
> We emphasize that the lineage example has **86,416 cells** on which UPFI runs in a **matter of minutes**.
>
> These timings show that UPFI runs for realistic $N$ and $d$ in short time. In comparison, DeepRUOT is an order of magnitude slower whilst less accurate. **We will add a discussion regarding the scalability and computational cost, as well as timing results, at the camera ready stage.**
>
> > Lack of ablations
>
> We add regularization ablations for bifurcating system in $d = 10$. For varying $\alpha$ and $\lambda$ we show the cosine error for force recovery as well as Pearson correlation for growth rate recovery.
>
> ### Regularization ablation
> **Force recovery, cosine**
> ||$\lambda$=0.0|0.01|0.1|
> |-:|-:|-:|-:|
> |$\alpha$=0|0.104997|0.059309|0.0782117|
> |0.01|0.104997|0.0593571|0.0599831|
> |0.1|0.104997|0.0856431|0.203179|
>
> **Growth rate recovery, Pearson**
>
> ||0.0|0.01|0.1|
> |-:|-:|-:|-:|
> |0|0.922827|0.91711|0.916789|
> |0.01|0.922827|0.916712|0.898708|
> |0.1|0.922827|0.909402|0.349477|
>
> For $\lambda = 0$ there is no regularization, and the error is larger than when $\lambda > 0$. From the growth rate perspective, no regularization performs better, while this incurs a larger error on the force field. From these measurements we find that the growth rate is less sensitive to the regularization. We interpret this as resulting from an **implicit regularization** arising from parameterisation with relatively small neural nets. The motivation for regularization arises from theoretical considerations to ensure uniqueness, as motivated by our Theorem. We can draw from these ablation study a simple rule of thumb for UPFI: use small or no regularization with moderately sized neural networks. **Based on this comment, we propose to add these ablations experiments and discuss their consequences on the architectures choices in the manuscript at the camera ready stage .**
>
> ### Additional comments
> > Velocity prior: Could kernel-smoothed RNA-velocity guide or evaluate the drift, as in [4]?
>
> Yes this could be done, however we find that RNA velocity is not reliable enough to be used directly since it requires strong biophysical assumptions to be inferred in the first place [Gorin2022]
>
> ```
> [Sejourne2019] Thibault Séjourné, Jean Feydy, Francois-Xavier Vialard, Alain Trouvé, and Gabriel Peyré. Sinkhorn divergences for unbalanced optimal transport.
> [Coomer2022] Coomer MA, Ham L, Stumpf MP. Noise distorts the epigenetic landscape and shapes cell-fate decisions. Cell Systems
> [Eyring2023] Eyring L, Klein D, Uscidda T, Palla G, Kilbertus N, Akata Z, Theis F. Unbalancedness in neural monge maps improves unpaired domain translation. arXiv preprint arXiv:2311.15100. 2023 Nov 25.
> [Tong2023] Tong A, Malkin N, Fatras K, Atanackovic L, Zhang Y, Huguet G, Wolf G, Bengio Y. Simulation-free schr\" odinger bridges via score and flow matching. arXiv preprint arXiv:2307.03672. 2023 Jul 7.
> [Gorin2022] Gorin G, Fang M, Chari T, Pachter L. RNA velocity unraveled.
> ```

---

> > ### Comment · Reviewer_rQy3 · 2025-08-04
> >
> > Thank you for providing new ablations, results for higher dimensions, and provide further positioning of the paper. This rebuttal has addressed most of my concerns, and ready to revise my score to accept.

---

> > > ### Author Response · Authors · 2025-08-05
> > >
> > > Thank you again for your time reading our paper and for your feedback. We are glad to hear that we most of your concerns have been addressed in our rebuttal, and we commit to implementing these changes in the final paper. We are appreciative of your revision of the original score.

---

### Official Review · Reviewer_wbX3 · 2025-07-02

**Clarity:** 4
**Significance:** 2
**Originality:** 2
**Rating:** 4
**Confidence:** 3

**Summary:**

The manuscript focuses on the task of predicting single-cell trajectories and inferring their underlying dynamics. The authors adapt the PFI approach to the unbalanced case, allowing the model to capture both cell growth and cell death. They present results on both toy and single-cell datasets.

**Questions:**

- Lines 60–61: Should one of the conditions be g(x) < 0.
- Could you provide more intuition for Equation (12)? For example, it seems that as alpha tends to zero, it converges to the dynamic OT formulation.
- Is the model always an OU process with quadratic fitness? This is not entirely clear from Algorithm 1.
- In Figure 4 (ii), if you rank the top-k elements of matrix A, how many correspond to true interactions (the ones shown in red)? Reporting this would help compare the two learned matrices more easily.
- Is the algorithm robust to the “Sinkhorn loss on marginals” step in Algorithm 1? Have you experimented with different solvers for this step? Specifically, does this step require an unbalanced Sinkhorn solver?

**Ethical Concerns:**

["NO or VERY MINOR ethics concerns only"]

**Final Justification:**

Thank you for engaging in the discussion, the rebuttal answers my questions and addresses the concerns.

**Limitations:**

A more detailed discussion of limitations should be included, or they should be highlighted through experiments—for example, the computational cost associated with using a neural ODE.

**Paper Formatting Concerns:**

no concerns

**Quality:**

3

**Strengths And Weaknesses:**

Overall, the paper and method are clear and easy to follow. The approach is well motivated by biological considerations—such as cellular growth and death—that are sometimes omitted in other methodologies. The authors propose an unbalanced version of PFI, which demonstrates improvements over the original PFI on both synthetic and real datasets. They also present a theoretical contribution by showing the existence of a unique minimizer for the continuous version of their loss function.

However, I believe comparisons against flow matching-based methods are missing. For instance, it would be informative to include a vanilla flow matching model or even an unbalanced one (akin to [1]).

[1] Eyring, L., Klein, D., Uscidda, T., Palla, G., Kilbertus, N., Akata, Z., & Theis, F. (2023). Unbalancedness in neural Monge maps improves unpaired domain translation. arXiv preprint arXiv:2311.15100.

---

> ### Author Rebuttal · Authors · 2025-07-31
>
> We thank the reviewer for their thorough review. Below we adress the reviewer's comments regarding i) the scalability of our approach and its comparison to Flow Matching, ii) the choice of the regularization, and we clarify a number of points.
>
> ### Scalability and comparison to Flow Matching
> > A more detailed discussion of limitations should be included, or they should be highlighted through experiments—for example, the computational cost associated with using a neural ODE.
>
> We agree with the reviewer that in its current form the paper misses a discussion about the scalability and computational cost of the approach. Concerning the scalability, we mitigate the neural ODE cost by only taking 2 to 3 Euler steps in between snapshots. This trades off accuracy for scalability. The unbalanced Sinkhorn loss benefits from a good sample complexity (see [Sejourne2019] and our response to Reviewer `wcim`). For the scales of data available in typical time-resolved single-cell RNA-seq datasets ($\leq 10^5$ cells with $\sim 5$ time-points), we can scale our approach to moderately high dimensions of ~50. To demonstrate this, we run the bifurcating example (Table 1 of our paper) with 25 and 50 dimensions.
>
> We also implement OT-conditional flow matching (OTFM) [Tong2023] and an unbalanced variant (UOTFM) [Eyring2023].
>
> ### Bifurcating ex. (Table 1) to $d = 25, 50$
>
> **Cosine error**
> |$d$|UPFI|PFI|ODE|TIGON|DeepRUOT|OTFM|UOTFM|
> |-------:|:--------------------|:--------------------|:----------------|:----------------|:------------------|:----------------|:----------------|
> |25|**0.06±0.00**|0.09±0.00|0.19±0.04|0.27±0.01|0.37±0.00|0.60±0.01|0.63±0.02|
> |50|**0.06±0.00**|0.07±0.00|0.26±0.03|0.25±0.01|0.44±0.00|0.55±0.01|0.52±0.01|
>
> **Fate prob.**
> |$d$|UPFI|PFI|ODE|TIGON|DeepRUOT|OTFM|UOTFM|
> |---:|:--------------------|:----------------|:----------------|:----------------|:----------------|:----------------|:----------------|
> |25|**0.97±0.01**|0.65±0.02|0.90±0.02|0.77±0.01|0.95±0.00|0.74±0.02|0.93±0.01|
> |50|**0.98±0.00**|0.63±0.03|0.91±0.01|0.80±0.02|0.92±0.00|0.73±0.01|0.93±0.01|
>
>
> We can see that while UPFI still outperforms the other methods, UOTFM does a decent job at recovering the fate probabilities, certainly because it handles appropriately the unbalancedness of the two branches. We also computed the timings for our method on the different experiments we ran, and we see that the force training and score training take a similar amount of time, which are very reasonable:
>
> ### Runtime (in seconds, for Table 1)
>
> |dim|Score|UPFI|DeepRUOT|
> |------:|:------------------|:------------------|-------------:|
> |2|61.32±0.50|86.83±0.62|1156.21|
> |5|61.21±0.43|86.99±0.38|1036.71|
> |10|61.26±0.42|87.22±0.51|1002.07|
> |25`new!`|249.54±1.00|442.58±2.01|1194.7|
> |50`new!`|249.25±0.34|440.84±1.87|1079.56|
>
> Note we add results for $d=25,50$, for which we increase score matching iters from 25k to 100k, and UPFI iters from 5k to 25k.
>
> ### Runtime (in seconds, for Fig.5)
>
> |Score|UPFI|
> |----------:|----------:|
> |24.19±0.13|207.97±1.63|
>
> Compared to UPFI, DeepRUOT is much slower to train and is less accurate.
>
> **We will discuss more thoroughly these scaling issues and include these additional experiments in the paper at the camera ready stage.**
>
> > However, I believe comparisons against flow matching-based methods are missing. For instance, it would be informative to include a vanilla flow matching model or even an unbalanced one (akin to [1]).
>
> We agree with the reviewer that we lack a proper comparison with Flow Matching approaches. We performed additional numerical experiments to fill in this gap (see response to previous question). For the bifurcating example they are reported as the last two last columns of the table, and they confirm that although performing comparably to other methods, balanced and unbalanced flow matching (resp. OTFM and UOTFM) still underperform with respect to UPFI. We would like to stress that UPFI, although not as scalable as Flow Matching, allows to jointly infer drift and growth, while simultaneously accounting for multiplicative (state and drift dependent) noise. Such models, which are biologically meaningful are not currently supported in existing Flow Matching frameworks. **We will include these new results at the camera ready stage, as well as to improve the literature review of flow matching approaches. This will also be an opportunity to clarify the positioning of our approach, and what sets it apart from these more scalable methods.**
>
> ### Model and regularization choices
> > Could you provide more intuition for Equation (12)? For example, it seems that as alpha tends to zero, it converges to the dynamic OT formulation.
>
> The regularizer is the Wasserstein-Fisher-Rao energy, as introduced in [Chizat2018]. This distance generalizes the Benamou-Brénier dynamic formulation of optimal transport to unbalanced optimal transport. Overall, our loss function includes a data fitting constraint via the Sinkhorn divergences - which can be seen as a soft penalty on the marginals at all times points - and a WFR loss in between time points. With this choice of regularizer, our loss resembles an approximation of a multi-marginal unbalanced optimal transport. However, it is important to note that our method diverges from unbalanced optimal transport since we do not penalize the phase-space velocity (which for Fokker-Planck flows has a contribution stemming from the score, see Eq. 3 or Eq. 4), but only the force field. In practice, because of concerns regarding the identifiability problem, we also enforce the force field and growth rate to be autonomous. This additionally differentiates our approach from dynamic unbalanced optimal transport.
>
> > Is the model always an OU process with quadratic fitness? This is not entirely clear from Algorithm 1.
>
> The OU/quadratic process is a toy system for which we can provide theoretical analysis. Alg. 1 applies in the general (i.e. non-linear) setting. **We will clarify the distinction at the camera ready state.**
>
>
> ### Additional comments
> > Is the algorithm robust to the “Sinkhorn loss on marginals” step in Algorithm 1? Have you experimented with different solvers for this step? Specifically, does this step require an unbalanced Sinkhorn solver?
>
> Yes the loss functional on the marginals needs to be able to handle positive (empirical) measures with differing mass. While different losses can be used (and not necessarily unbalanced OT, see e.g. [Mroueh2020]), we opt for UOT as a natural choice. In practice, we use the GeomLoss implementation which uses Sinkhorn-type iterations.
>
> > In Figure 4 (ii), if you rank the top-k elements of matrix A, how many correspond to true interactions (the ones shown in red)? Reporting this would help compare the two learned matrices more easily.
>
> We thank the reviewer for this spot on suggestion. This is what the precision-recall curves shown in Fig.4(iii) measures as a **summary over all possible thresholds**. Given a threshold value, we identify the entries in the matrix that are larger than this threshold. Among these thresholded entries, the precision computes the proportion of true interactions. The recall on the other hand computes the proportion of true interactions recovered among all true interactions in the unthresholded matrix. A large recall and large precision are synonym of good prediction. As the threshold level is not prescribed, we vary it to build the precision-recall curve, and compute the area under the curve. A larger area under the curve indicates better recovery of the underlying true interactions.
>
> ```
> [Mroueh2020] Mroueh Y, Rigotti M. Unbalanced sobolev descent. Advances in Neural Information Processing Systems. 2020;33:17034-43.
> [Chizat2018] Lenaic Chizat and Francis Bach. On the global convergence of gradient descent for over- parameterized models using optimal transport. In Advances in neural information processing systems, pages 3036–3046, 2018
> [Eyring2023] Eyring L, Klein D, Uscidda T, Palla G, Kilbertus N, Akata Z, Theis F. Unbalancedness in neural monge maps improves unpaired domain translation. arXiv preprint arXiv:2311.15100. 2023 Nov 25.
> [Tong2023] Tong A, Malkin N, Fatras K, Atanackovic L, Zhang Y, Huguet G, Wolf G, Bengio Y. Simulation-free schr\" odinger bridges via score and flow matching. arXiv preprint arXiv:2307.03672. 2023 Jul 7.

---

> > ### Comment · Reviewer_wbX3 · 2025-08-04
> >
> > Thank you for your answer and clarifications.
> >
> > I appreciate the new comparison with flow matching, could you also add it to the runtime experiment ? It may do better, but at least the reader can understand the tradeoff between speed and accuracy.

---

> > > ### Author Response · Authors · 2025-08-04
> > >
> > > Thank you again for your feedback and suggestions on our paper, and we appreciate your time spent reviewing our work .
> > > We completely agree that it is important to highlight the tradeoff between speed and accuracy between our (and other ODE-based methods) and flow matching.
> > >
> > > Below we show results for $d = 25, 50$ and for ease of reference we show the accuracy metrics we reported earlier in our rebuttal and runtimes side-by-side, all results are shown across 5 seeds. As expected, flow matching runs much faster (all methods trained for 25k iterations, and score model trained for 100k iterations) but is much less accurate compared to UPFI.
> > >
> > > We commit to adding comprehensive timing results in the final version of our paper.
> > >
> > > ### Accuracy
> > >
> > > **Cosine error**
> > > |$d$|UPFI|PFI|ODE|TIGON|DeepRUOT|OTFM|UOTFM|
> > > |-------:|:--------------------|:--------------------|:----------------|:----------------|:------------------|:----------------|:----------------|
> > > |25|**0.06±0.00**|0.09±0.00|0.19±0.04|0.27±0.01|0.37±0.00|0.60±0.01|0.63±0.02|
> > > |50|**0.06±0.00**|0.07±0.00|0.26±0.03|0.25±0.01|0.44±0.00|0.55±0.01|0.52±0.01|
> > >
> > > **Fate prob.**
> > > |$d$|UPFI|PFI|ODE|TIGON|DeepRUOT|OTFM|UOTFM|
> > > |---:|:--------------------|:----------------|:----------------|:----------------|:----------------|:----------------|:----------------|
> > > |25|**0.97±0.01**|0.65±0.02|0.90±0.02|0.77±0.01|0.95±0.00|0.74±0.02|0.93±0.01|
> > > |50|**0.98±0.00**|0.63±0.03|0.91±0.01|0.80±0.02|0.92±0.00|0.73±0.01|0.93±0.01|
> > >
> > > ### Runtime (s)
> > >
> > > |   dim | Score matching           | UPFI            | PFI             | ODE              | TIGON           | OTFM            | UOTFM          |
> > > |------:|:------------------|:------------------|:------------------|:-------------------|:------------------|:------------------|:-----------------|
> > > |    25 | 249.73 $\pm$ 0.50 | 439.48 $\pm$ 1.51 | 333.51 $\pm$ 2.02 | 861.41 $\pm$ 11.43 | 313.79 $\pm$ 3.71 | 100.68 $\pm$ 0.84 | 52.64 $\pm$ 0.91 |
> > > |    50 | 250.71 $\pm$ 1.52 | 443.52 $\pm$ 2.23 | 335.94 $\pm$ 3.65 | 868.93 $\pm$ 11.86 | 316.18 $\pm$ 4.17 | 99.61 $\pm$ 0.68  | 52.95 $\pm$ 0.75 |

---

### Official Review · Reviewer_wcim · 2025-07-07

**Clarity:** 4
**Significance:** 3
**Originality:** 3
**Rating:** 5
**Confidence:** 3

**Summary:**

The paper looks at the challenge of reconstructing dynamical systems from snapshots. In particular, for single cell data, the sample is destroyed by the measurement. The authors present "unbalanced" probability flow inference. Which has somewhat of the feel of a method for dealing with censored data that takes account of the underlying dynamics. The paper modifies the probability flow inference technique to deal with a limitation where proliferation and death isn't handled.

**Questions:**

I would like to hear more about alorithm scaling. My sense is currently this approach might be hard to apply to some data sets.

Is there an example where growth is critical and current methods cannot be deployed as a result? Even if it's a motivational example it would be good to hear it.

I've been out of the loop for comp bio for a bit, and we were discussing snapshot data some years ago. I was a little surprised there wasn't a review of progress in that direction that goes beyond listing references.

**Ethical Concerns:**

["NO or VERY MINOR ethics concerns only"]

**Limitations:**

yes

**Quality:**

3

**Strengths And Weaknesses:**

The paper does a nice job of explaining the problem, being clear about what previous work does and where the contribution has come in.

The algorithm is illustrated with a range of experiments that help both in understanding of the set up and the quality of results.

Formatting and description of algorithm is a good balance between technical detail and clarity of explanation

The problem in single cell data is well motivated and the method is compared with other methods from the literature and performs in the manner that would be expected given enhancements.

Experimental set up is clearly explained and easy to follow. Sensible simulations are given before the full experiments.

In an ideal world we would have seen more from real RNA-seq data or other data sets with similar problems.

I suspect the use of 10 principal components is due to challenges in scaling the algorithm, I would have liked to hear more about this.

---

> ### Author Rebuttal · Authors · 2025-07-31
>
> We thank the reviewer for their appreciation of our efforts to provide a clear and mathematically rigorous approach. Below we address the different reviewer's comments regarding i) the scalability of the approach, ii) the motivation for the inference of growth.
>
> ### Scalability
> > I suspect the use of 10 principal components is due to challenges in scaling the algorithm, I would have liked to hear more about this.
> > I would like to hear more about alorithm scaling. My sense is currently this approach might be hard to apply to some data sets.
>
> **Theoretical complexity**
>
> The unbalanced Sinkhorn divergence can be computed efficiently with per-iteration complexity $\mathcal{O}(B^2)$, with a dimension-independent sample complexity $\mathcal{O}(B^{-1/2})$ [Sejourne2019], provided $B$ is large enough, where $B$ is the batch size. The denoising score matching on the other hand has a per-iteration complexity $\mathcal{O}(B\times d)$ where $d$ is the dimension. For the ODE integration, we can mitigate the complexity at the price of accuracy by simpling taking less Euler steps in between time points. In practice we find that taking 2 to 3 steps is satisfying. While this setting still prevents us from working with thousands of dimensions, it isn't preventing us from scaling to ~50 dimensions, as we illustrate with additional results on the bifurcating example. Note the addition of (U)OTFM (flow matching), as per request of `wbX3` and `rQy3`.
>
> ### Bifurcating ex. (Table 1) to $d = 25, 50$
>
> **Cosine error**
> |$d$|UPFI|PFI|ODE|TIGON|DeepRUOT|OTFM|UOTFM|
> |-------:|:--------------------|:--------------------|:----------------|:----------------|:------------------|:----------------|:----------------|
> |25|**0.06±0.00**|0.09±0.00|0.19±0.04|0.27±0.01|0.37±0.00|0.60±0.01|0.63±0.02|
> |50|**0.06±0.00**|0.07±0.00|0.26±0.03|0.25±0.01|0.44±0.00|0.55±0.01|0.52±0.01|
>
> **Fate prob.**
> |$d$|UPFI|PFI|ODE|TIGON|DeepRUOT|OTFM|UOTFM|
> |---:|:--------------------|:----------------|:----------------|:----------------|:----------------|:----------------|:----------------|
> |25|**0.97±0.01**|0.65±0.02|0.90±0.02|0.77±0.01|0.95±0.00|0.74±0.02|0.93±0.01|
> |50|**0.98±0.00**|0.63±0.03|0.91±0.01|0.80±0.02|0.92±0.00|0.73±0.01|0.93±0.01|
>
> **Use of PCA**
> This has consequences for the analysis of real biological datasets like Hematopoietic Stem Cells differentiation, for which only a few tens of critical transcription factor are responsible for the differentiation of cells [Weinreb2020]. On the Lineage Tracing dataset, using additional lineage barcoding information, it was shown that there exists at least ten genes whose expression levels are better predictors of fate probability than any state-of-the-art algorithms working in reduced dimensional PCA space [Weinreb2020]. For this reason we believe that the scaling limitations of UPFI do not completely hinder its predictive potential for the prediction of cell fate decision making, since the difficulty seemingly resides more in building a relevant list of genes or a better dimensional reduction method than PCA. **Motivated by this comment, we will to include a discussion about the scalability and computational cost at the camera stage.**
>
> **Practical runtime**
>
> On the practical front, we find that UPFI is **as fast or faster to run** compared to competing methods. Below we provide runtimes (in seconds, over 5 repeats) for the bifurcating and lineage (Figs 3,5 of our paper) and compare to DeepRUOT.
>
> ### Runtime (in seconds, for Table 1)
>
> |dim|Score|UPFI|DeepRUOT|
> |------:|:------------------|:------------------|-------------:|
> |2|61.32±0.50|86.83±0.62|1156.21|
> |5|61.21±0.43|86.99±0.38|1036.71|
> |10|61.26±0.42|87.22±0.51|1002.07|
> |25`new!`|249.54±1.00|442.58±2.01|1194.7|
> |50`new!`|249.25±0.34|440.84±1.87|1079.56|
>
> Note we add results for $d=25,50$, for which we increase score matching iters from 25k to 100k, and UPFI iters from 5k to 25k.
>
> ### Runtime (in seconds, for Fig.5)
>
> |Score|UPFI|
> |----------:|----------:|
> |24.19±0.13|207.97±1.63|
>
> We emphasize that the lineage example has 86,416 cells on which UPFI runs in a matter of a few minutes.
>
> ### Motivation for the growth model
> > Is there an example where growth is critical and current methods cannot be deployed as a result? Even if it's a motivational example it would be good to hear it.
>
> We thank the reviewer for raising the issue of the necessity of a growth model, this is an important point which we didn't elaborate enough upon. In our cellular differentiation example we infer that growth is decreasing with time, with progenitor cells and stem cells dividing more than terminal cell types (Monocytes and Neutrophiles). While this is expected [Sha2024], this indicates that growth has a rather limited impact on the inferred drift since both terminal states which emerge at a similar time in the differentiation process have comparable growth rates. We observe however that taking growth into account improves by a factor ~3 the cell fate prediction accuracy with respect to PFI. A more robust and precise quantification of the role of the growth rate would be necessary to understand why taking the growth rate into account improves the prediction. This is work we plan to do, but which goes beyond this method-oriented paper.
>
> We can however provide examples where growth plays a clear role. Such a situation, which happens when growth is heterogeneous over cell types, is illustrated on our toy bifurcating example. In real systems, it was observed when attempting to learn Optimal Transport trajectories summarizing a cell reprogramming experiment in [Schiebinger2019]. In this experiment, differentiated cells where reprogrammed towards induced pluripotent stem cells (iPSCs) via an experimental protocol. During the course of the reprogramming, intermediate stromal cell expressing apoptotic genes (inducing cell death) emerged. The misinterpretation of the death of these stromal cells as a transition to another cell state (transport) lead to incorrect reprogramming trajectories connecting them to the iPSCs emerging at the end of the reprogramming experiment. This mistake could only be corrected by accounting for cell death in the dynamical model. **To motivate our method, we propose to include this example in the introduction of the paper.**
>
>
> ### Additionnal comments
> > I've been out of the loop for comp bio for a bit, and we were discussing snapshot data some years ago. I was a little surprised there wasn't a review of progress in that direction that goes beyond listing references.
>
> The reviewer is right that a more in-depth discussion of related work is warranted. In this instance we were not able to provide this due to the page limit, and we provided an in-depth discussion of TIGON and DeepRUOT in the appendix. **We propose to rework the introduction to include an improved related work section.**
>
>
> ```
> [Sejourne2019] Thibault Séjourné, Jean Feydy, Francois-Xavier Vialard, Alain Trouvé, and Gabriel Peyré. Sinkhorn divergences for unbalanced optimal transport. arXiv preprint arXiv:1910.12958, 2019.
> [Weinreb2020] Caleb Weinreb, Alejo Rodriguez-Fraticelli, Fernando D Camargo, and Allon M Klein. Lineage tracing on transcriptional landscapes links state to fate during differentiation. Science, 367(6479):eaaw3381, 2020.
> [Schiebinger2019] Geoffrey Schiebinger, Jian Shu, Marcin Tabaka, Brian Cleary, Vidya Subramanian, Aryeh Solomon, Joshua Gould, Siyan Liu, Stacie Lin, Peter Berube, et al. Optimal-transport analysis of single-cell gene expression identifies developmental trajectories in reprogramming. Cell, 176(4):928–943, 2019.
> [Sha2024] Yutong Sha, Yuchi Qiu, Peijie Zhou, and Qing Nie. Reconstructing growth and dynamic trajectories from single-cell transcriptomics data. Nature Machine Intelligence, 6(1):25–39, 2024.
> ```

---

> > ### Comment · Reviewer_wcim · 2025-08-04
> > **Thanks for the clarifications**
> >
> > I think these ideas would significantly strengthen what is already a very interesting paper.

---

> > > ### Author Response · Authors · 2025-08-04
> > >
> > > Thank you again for your time spent on our paper and positive evaluation. Indeed we agree that incorporating these additions would strengthen our paper and we commit to incorporating these additions in the final version.

---

### Note · Authors · 2025-08-13

We thank all reviewers for their detailed and thoughtful feedback on our work, and the AC and SAC for handling our submission. Thanks to reviewer feedback, we have been able to strengthen our work in several directions. These additions and clarifications have been well received, as evidenced by one reviewer raising their score and the remaining two reviewers maintaining their positive score.

We will incorporate all of the material from our rebuttal into the final version of our paper, including:
* Improvement of the discussion of related work, in particular to provide a discussion comparing our approach to flow matching methods [reviewers wcim, wbX3, rQy3]
* Discussion of the scalability and computational cost of our method, including runtimes [reviewers wcim, rQy3]
* Added results for bifurcating example (Table 1) for $d = 25, 50$ as well as comparing to (un)balanced flow matching [reviewers wcim, wbX3, rQy3].
* Further discussion and details on the motivation for growth model [reviewer wcim]
* Discussion of choice of regularization parameters and ablation results [reviewer rQy3]

We have also fixed various issues with referencing and formatting, including those pointed out by reviewers.

Once again, thank you all for your engagement and input in this process thus far.

The Authors

---

### Decision · Program_Chairs · 2025-09-17

**Decision:**

Accept (poster)

**Comment:**

This paper studies a multimarginal stochastic transport problem with applications to various simulated systems and to single-cell RNA sequencing time series data. The transport problem studied allows for division and death coefficient, a model that is more suitable for some applications than the measure-preserving models studied in most past work. It is solved using a simulation-based algorithm that minimises Sinkhorn divergences between target marginals provided by samples and simulated marginals pushed forward from a previous time step.

All reviewers are positive about the paper. They particularly praise the solid technical contribution (including theoretical guarantees of uniqueness of the solution), the clarity of exposition, and the quality of the results.

Main weaknesses:
- Concerns about scaling to higher dimennsion and computation cost (wcim, wbX3, rQy3). This was answered in the rebuttal by new experiments that show the results hold up when the dimensionality of synthetic bifurcating data is increased from 10 to 50 and timing results. The authors are encouraged to include these in the final version and to study the scaling of the biological data experiments with more PCA components as well, which would answer wcim's concerns more fully.
- Review of related work on inference of dynamics from snapshot data (wcim, rQy3). The authors promised extended discussion in the revision.
- Lack of comparison with simulation-free dynamic OT methods such as flow matching (wbX3, rQy3). New experiments added in the rebuttal answered this concern.

I follow the reviewers' recommendation and recommend acceptance.